# The ataxin-1 interactome reveals direct connection with multiple disrupted nuclear transport pathways

Sunyuan Zhang[1], Nicholas A. Williamson [2], Lisa Duvick[3], Alexander Lee[4], Harry T. Orr[3], Austin Korlin-Downs[3], Praseuth Yang[3], Yee-Foong Mok[2], David A. Jans [4✉] & Marie A. Bogoyevitch[1]

The expanded polyglutamine (polyQ) tract form of ataxin-1 drives disease progression in spinocerebellar ataxia type 1 (SCA1). Although known to form distinctive intranuclear bodies, the cellular pathways and processes that polyQ-ataxin-1 influences remain poorly understood. Here we identify the direct and proximal partners constituting the interactome of ataxin-1[85Q] in Neuro-2a cells, pathways analyses indicating a significant enrichment of essential nuclear transporters, pointing to disruptions in nuclear transport processes in the presence of elevated levels of ataxin-1. Our direct assessments of nuclear transporters and their cargoes confirm these observations, revealing disrupted trafficking often with relocalisation of transporters and/or cargoes to ataxin-1[85Q] nuclear bodies. Analogous changes in importin-β1, nucleoporin 98 and nucleoporin 62 nuclear rim staining are observed in Purkinje cells of *ATXN1*[82Q] mice. The results highlight a disruption of multiple essential nuclear protein trafficking pathways by polyQ-ataxin-1, a key contribution to furthering understanding of pathogenic mechanisms initiated by polyQ tract proteins.

---

[1] Department of Biochemistry and Molecular Biology, University of Melbourne, Parkville, VIC 3010, Australia. [2] Bio21 Molecular Science and Biotechnology Institute, University of Melbourne, Parkville, VIC 3010, Australia. [3] Institute of Translational Neuroscience, and Department of Laboratory Medicine and Pathology, University of Minnesota, Minneapolis, MN 55455, USA. [4] Nuclear Signalling Lab., Department of Biochemistry and Molecular Biology, Monash University, Clayton, VIC 3800, Australia. ✉email: david.jans@monash.edu

Spinocerebellar ataxia type 1 (SCA1) is an autosomal dominant neurodegenerative disease, associated with disabilities in coordination and movement and a marked atrophy in the cerebellum and brainstem[1]. The genetic cause underlying SCA1 has been mapped to *ATXN1*, the gene encoding the ataxin-1 protein, whereby the CAG nucleotide repeat region of *ATXN1* is expanded in SCA1 patients[2]. The resulting polyglutamine (polyQ) tract form of the ataxin-1 protein, polyQ-ataxin-1, forms distinctive nuclear bodies in individuals with SCA1, a feature recapitulated in SCA1 transgenic mice[3]. More broadly, polyQ tract expansions in specific proteins are now appreciated to drive at least 10 diseases[4], with their study providing exciting insights that extend beyond critical aspects of protein biochemistry including protein folding/misfolding and protein-protein interactions to important points of regulation in cellular homoeostasis dictated by proteostasis and impacts on cell survival/death-decision making[5,6].

Disruptions in nuclear import/export processes in neurodegenerative diseases such as amyotrophic lateral sclerosis (ALS), Huntington's disease (HD), and Alzheimer's Disease (AD) have been reported[7–11]. The nuclear transport machinery responsible for the regulated trafficking of proteins between the cytoplasm and the nucleus has a number of key components: nuclear import/export signals (NLS/NES) of the cargo proteins that direct their nuclear/cytoplasmic distributions, dedicated transport proteins responsible for nuclear import (importins) and export (exportins), the nuclear pore complex that spans the nuclear envelope and provides a regulated gateway for nuclear trafficking events, and the RanGTP/RanGDP system that drives directionality of the transport events[12]. Disrupting any of these entities can influence nucleocytoplasmic trafficking[13], making each of these a potential player in altered nuclear trafficking in neurodegenerative disease.

Here we approach the actions of polyQ-ataxin-1 from the perspective of protein interaction networks, by analysing the interactome of ataxin-1[85Q]. By combining direct interaction analyses with our identification of polyQ-ataxin-1 proximal partners[14] and pathways analyses, we not only confirm known ataxin-1 interacting partners but we identify the nuclear transport pathway as the top-ranked cellular process defined by these interactors. By interacting with or sequestering additional proteins into ataxin-1 nuclear bodies, polyQ-ataxin-1 has the potential to disrupt cellular homoeostasis[15–17]. To assess this possibility of ataxin-1 driven nuclear transport disruption, we define an immediate disruption of the localisation of multiple components of the nuclear transport machinery, often with their mis-localisation to ataxin-1[85Q] nuclear bodies in cells transiently expressing polyQ-ataxin-1. Moreover, we extend these observations to demonstrate altered nuclear transport machinery in a SCA1 mouse model that develops symptoms of ataxia arising from the expression of the pathological form of polyQ-ataxin-1. Our results reinforce a disruption of nuclear transport as contributing to the impact of polyQ-ataxin-1.

## Results and discussion

**Ataxin-1[85Q] nuclear bodies are enhanced by arsenite stress.** To define the suitability of the polyQ-ataxin-1 constructs GFP-ataxin-1[85Q] and MBI-ataxin-1[85Q] for interactome analyses in Neuro-2a cells, we assessed the subcellular localisation of these proteins, compared to GFP or MBI alone, under control conditions and further in response to the pro-oxidant stressor arsenite[18,19]. Whilst GFP remains broadly distributed throughout the cell, GFP-ataxin-1[85Q] forms distinctive nuclear bodies that have been reported across a range of cell types[15,20]. These GFP-ataxin-1[85Q] nuclear bodies formed within 24 h of transfection

and increased in size upon acute arsenite exposure (300 μM, 1 h) (Fig. 1a), a potent pro-oxidant stress reported previously for driving the formation of stress granules[21]. We further quantitated the significantly increased size of these GFP-ataxin-1[85Q] nuclear bodies upon arsenite exposure, demonstrating that these changes were not observed for the non-pathogenic GFP-ataxin-1 [85Q] S766A phosphorylation site mutant[22,23] (Supplementary Fig. 1a,b), and that live imaging presented evidence of nuclear body fusion events for GFP-ataxin-1[85Q] (Supplementary Fig. 1c). Notably, our assessment of ataxin-1 aggregate formation by analytical ultracentrifugation analyses of lysates prepared from these cells, showed increased aggregates for both GFP-ataxin-1 [85Q] and the non-pathogenic GFP-ataxin-1[85Q] S766A (Supplementary Fig. 1d), emphasising that aggregation state and visible body size do not always change in parallel as has been reported for the huntingtin protein with an expanded polyQ tract[24,25] and as we have recently observed for polyQ-ataxin-1[26]. These ataxin-1 nuclear bodies were distinct from nucleoli, as defined by co-staining for nucleolin that remained largely unchanged in size or morphology in the presence of ataxin-1 expression or arsenite exposure (Supplementary Fig. 2a); quantitative evaluation confirmed this lack of ataxin-1/nucleolin co-localisation with low (<−0.3) Pearson's correlation coefficient values (Supplementary Fig. 2b). Thus, the nuclear bodies formed by polyQ-ataxin-1 were distinctive structures responsive to altered environmental conditions.

Proximity biotinylation as driven by BirA* biotin ligase provides a proteomics strategy for proximal partner identification[27]. MBI-ataxin-1[85Q], the myc-tagged BirA* N-terminal fusion with ataxin-1[85Q], was thus validated as per our workflow (Fig. 1b). In contrast to MBI alone that was primarily cytoplasmic, MBI-ataxin-1[85Q] was largely restricted to the nucleus (Fig. 1c). This localisation was consistent with the nuclear localisation of GFP-ataxin-1[85Q] and the previous observation that nuclear localisation of ataxin-1 is important for its toxicity[28], albeit that MBI-ataxin-1[85Q] formed less distinctive nuclear bodies than that observed for GFP-ataxin-1[85Q] (Fig. 1c (anti-myc staining) vs Fig. 1a (GFP signal)). Thus, to assess the suitability of MBI-ataxin-1[85Q] further, we used streptavidin-based detection to detect biotinylation driven by the MBI-fusion constructs. We confirmed a widespread distribution of biotinylation throughout the cell nucleus and cytoplasm by MBI alone, whereas we observed that the MBI-ataxin-1[85Q]-driven biotinylation was largely localised to prominent nuclear bodies (Fig. 1c). Furthermore, we observed a more restricted biotinylation profile for MBI-ataxin-1[85Q] as defined by streptavidin-detection of biotinylated proteins separated by SDS-PAGE (Fig. 1d), consistent with interactions of the MBI-ataxin-1[85Q] dominated by those in a nuclear body context. Thus, GFP-ataxin-1[85Q] and MBI-ataxin-1[85Q] have the potential to provide independent information on the nuclear-interacting protein partners of ataxin-1[85Q].

**The ataxin-1 interactome includes nuclear transport proteins.** In subsequent analyses, we combined BioID proximity biotinylation (using MBI-ataxin-1[85Q]) and Pulldown protocols (using GFP-ataxin-1[85Q]) as powerful alternative and complementary approaches[29] to identify ataxin-1[85Q]-associating partner proteins. Thus, we employed both MBI-ataxin-1[85Q] and GFP-ataxin-1[85Q] in Neuro-2a cells under both control and arsenite stress conditions, choosing to integrate the data obtained these parallel qualitative approaches to define the polyQ-ataxin-1 interactome. Following trypsin-digestion of the proteins captured in the different isolation protocols, our mass spectrometry and peptide raw data analysis defined the proteins present in each

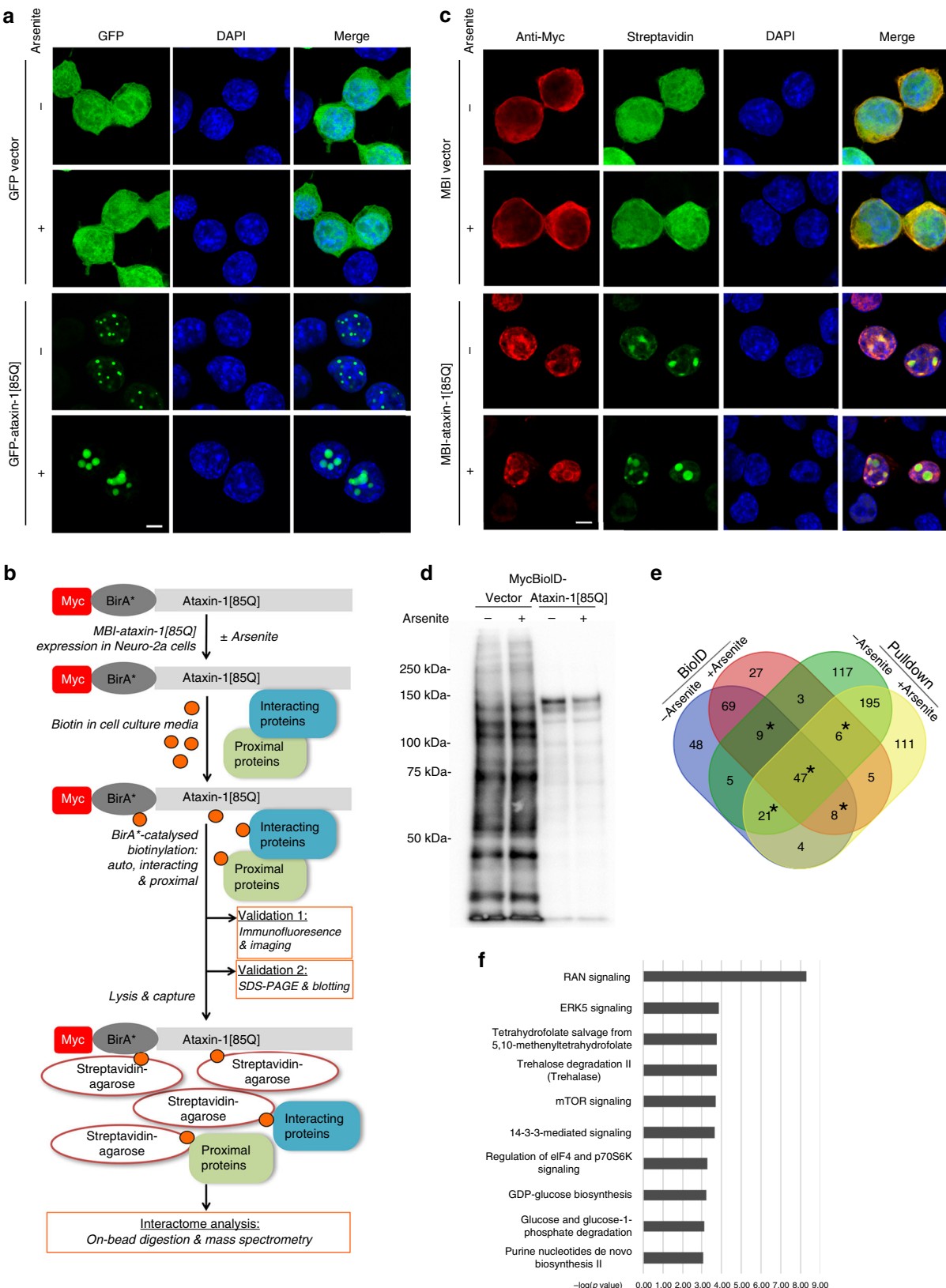

individual sample; for our analyses, we set the parameter that only proteins identified by >2 significant peptides would be retained for further analysis (Supplementary Fig. 3). From the samples analysed as biological triplicates across the four conditions (BioID ± arsenite, Pulldown ± arsenite), we removed proteins identified as background/non-specific (i.e. endogenous

biotinylation during incubation with biotin driven by MBI alone, or pulldown by GFP only, respectively). The result of this multistep analysis was a total of 675 proteins (Supplementary Data 1). This extends the breadth of the interactome of polyQ-ataxin-1, as previously investigated by proteins co-immunoprecipitating with myc-ataxin-1 in non-neuronal (HEK293T) cell lysates[30]. Our

**Fig. 1 Ataxin-1[85Q] interactome reveals enrichment of nuclear transport proteins.** Cells were transfected to express either GFP-vector or GFP-ataxin-1[85Q] (**a**) or MBI-vector or MBI-ataxin-1[85Q] with simultaneous addition of biotin (50 μM) (**b–d**). At 24 h post-transfection, cells were treated with arsenite (300 μM, 1 h) as indicated. Neuro-2a cells were then fixed and stained with DAPI (**a**), or fixed, stained with anti-myc antibody, Alexa 488-streptavidin and DAPI (**c**), before confocal laser scanning microscopy (CLSM) imaging. Representative images are shown; merge panels overlay GFP and DAPI images (**a**) or anti-myc, streptavidin and DAPI images (**c**), respectively. Scale bar = 10 μm. **b** Workflow for BioID sample validation and ataxin-1[85Q] interactome analysis in Neuro-2a cells. **d** At 24 h post-transfection, cells were lysed and the lysates subjected to SDS-PAGE, with biotinylated proteins subsequently detected using streptavidin-HRP. Results are typical of ≥ 3 independent experiments. Validated ataxin-1[85Q] constructs (in **a**, **c**, **d**) were used for ataxin-1[85Q] interactome identification by mass spectrometry. **e** Venn diagram overview of the 675 identified proteins from the four conditions grouped according to cell exposure (±arsenite treatment) and sample preparation (BioID or Pulldown). Areas marked by * indicate those groupings of proteins identified in ≥ 3 conditions. **f** Ingenuity Pathway Analysis (IPA) was performed on the 91 proteins shared by ≥ 3 conditions across biological triplicate samples. The statistically top-ranked category, as assessed by the –log($p$ value), was RAN Signalling that represents the proteins involved in protein nuclear transport.

qualitative data are available via ProteomeXchange, with identifier PXD010352[14]. Although additional quantitative proteomic analysis, for example by labelling sample proteins with isobaric tags followed by multiplexed analysis[31], would likely extend the interactome information to further include very low-abundance proteins or those showing variations in levels across the analysed samples, we identified ataxin-1 itself, as well as the well-characterised ataxin-1 partner protein, Capicua transcriptional repressor (CIC)[5], under all four conditions, supporting the robustness of our approaches.

In analysing the ataxin-1[85Q] interactome, we further refined the list of 675 proteins. Noting that the nucleolin protein was identified only by the BioID approach and thus may not show sustained interaction or co-localisation with ataxin-1[85Q], and that our initial characterisation clearly distinguished nucleoli from ataxin-1 nuclear bodies (Supplementary Fig. 2), we included only the 91 proteins observed in at least 3 out of 4 of the conditions analysed (Fig. 1e, indicated by *), thus ensuring their identification by both BioID and pull-down approaches. To statistically test for pathways showing over-representation relative to that expected by chance, we employed Ingenuity Pathway Analysis. This revealed the top-ranked pathway to be classified as "Ran signalling" (i.e. the proteins involved in the passage of proteins across the nuclear membrane) due to the inclusion of various proteins of the nuclear transport pathway, such as the importins which could directly allow NLS-dependent nuclear entry of the ataxin-1 protein[28], but also Ran-binding proteins and nucleoporins (Fig. 1f; see summary in Supplementary Table 1).

The length of the polyQ expansion in the ataxin-1 protein is known to influence the age-of-onset of disease symptoms in SCA1 patients[32]. A major consequence of polyQ length is the stabilisation of ataxin-1 protein that leads to higher ataxin-1 levels[33]; increasing levels of wild-type ataxin-1 can lead to SCA1-like neurodegeneration[34–36]. In the scenario of Pumilio haploin-sufficiency, even though many other targets are likely also affected, the upregulated wild-type ataxin-1 levels can lead to cerebellar neurodegeneration, as pathology could be largely corrected by reducing ataxin-1 levels[45]. Furthermore, reduced agility of young (5-week-old) animals of the unexpanded (ataxin-1[30Q]) A02/+ SCA1 transgenic model has been previously reported, as assessed by diminished rates of improvement on the accelerating rotarod apparatus and alterations in performance on the bar cross apparatus, despite no gross changes in cerebellar histology and rotarod performance in older (1-year-old) animals[37]. We thus also explored the possible influence of polyQ length acutely on the ataxin-1 interactome in our cultured cell system. We initially confirmed that comparable levels of ataxin-1[30Q] and ataxin-1[85Q] could cause comparable increases in cell death (Supplementary Fig. 4), stratifying our population analysis for levels of protein expression. Specifically, we noted that, in the majority of the cell population expressing either low (0.3–1 K

GFP fluorescence count) or moderate (1–10 K GFP fluorescence count) levels of ataxin-1 proteins, cell death was significantly increased to ~8%, whereas higher levels (10–100 K GFP fluorescence count) in a smaller subset of the cell population (~7%) could further cause significant increases in death to ~30% of that population subset. We then conducted parallel control experiments to identify the proximal partners for MBI-ataxin-1[30Q] or MBI-ataxin-1[85Q] in Neuro-2a cells under control and arsenite stress conditions. We performed analyses as per our BioID protocol (Supplementary Fig. 3b). Of the total of 455 proteins identified across these four conditions (from samples analysed as biological triplicates across the four conditions (ataxin-1[30Q] ± arsenite, ataxin-1[85Q] ± arsenite), and with the removal of non-specific proteins identified), 203 proteins were identified only for MBI-ataxin-1[30Q] whereas 30 proteins were identified only for MBI-ataxin-1[85Q] (Supplementary Fig. 3c; Supplementary Data 2). Notably, we again identified ataxin-1 itself, together with CIC and multiple proteins involved in nuclear transport (Supplementary Table 2) under these conditions.

We established the robustness of these identifications of partners for the ataxin-1[85Q] protein by employing a bimolecular fluorescence complementation (BiFC) assay that relies on protein interactions bringing together ectopically expressed Venus N-terminal (VN-) and Venus C-terminal (VC-) fragments to reconstitute fluorescence of the Venus protein, thus allowing direct visualisation of protein interactions in their normal cellular environment[38]. Importantly, BiFC analysis also allows detection of weak and transient interactions[39,40]. We first confirmed the robustness of this system by demonstrating that the known interacting pair of transcription factors, c-Fos and c-Jun, resulted in strong and homogeneous nuclear fluorescence consistent with previous studies[41], thus providing a robust positive control for this assay system; notably, in parallel testing of the c-Fos mutant, c-FosΔzip, paired with c-Jun, quantitative analysis confirmed nuclear fluorescence was significantly decreased consistent with the previous studies[41] and thus providing a robust negative control for this assay system (Fig. 2a).

Next, we used the interaction of ataxin-1[85Q] and the previously reported partner, CIC, as a further positive control for the BiFC system, as well as also testing the binding between ataxin-1[85Q] and our identified partner from the nuclear transport pathway, importin-α2. In evaluating the ataxin-1[85Q] + CIC pair, we observed distinctive nuclear puncta (Fig. 2b, upper panels), indicative of interaction primarily within the ataxin-1 nuclear bodies and consistent with the ataxin-1 localisation and MBI-ataxin-1[85Q]-driven biotinylation noted in Fig. 1. In parallel testing of the ataxin-1[85Q] V591A/S602D mutant (abbreviated herein as ataxin-1[85Q] mVS mutant) with known abrogated interaction with CIC that prevents SCA1 disease pathology[42], nuclear fluorescence was significantly decreased (Fig. 2b, upper panels). Thus, we conclude that this

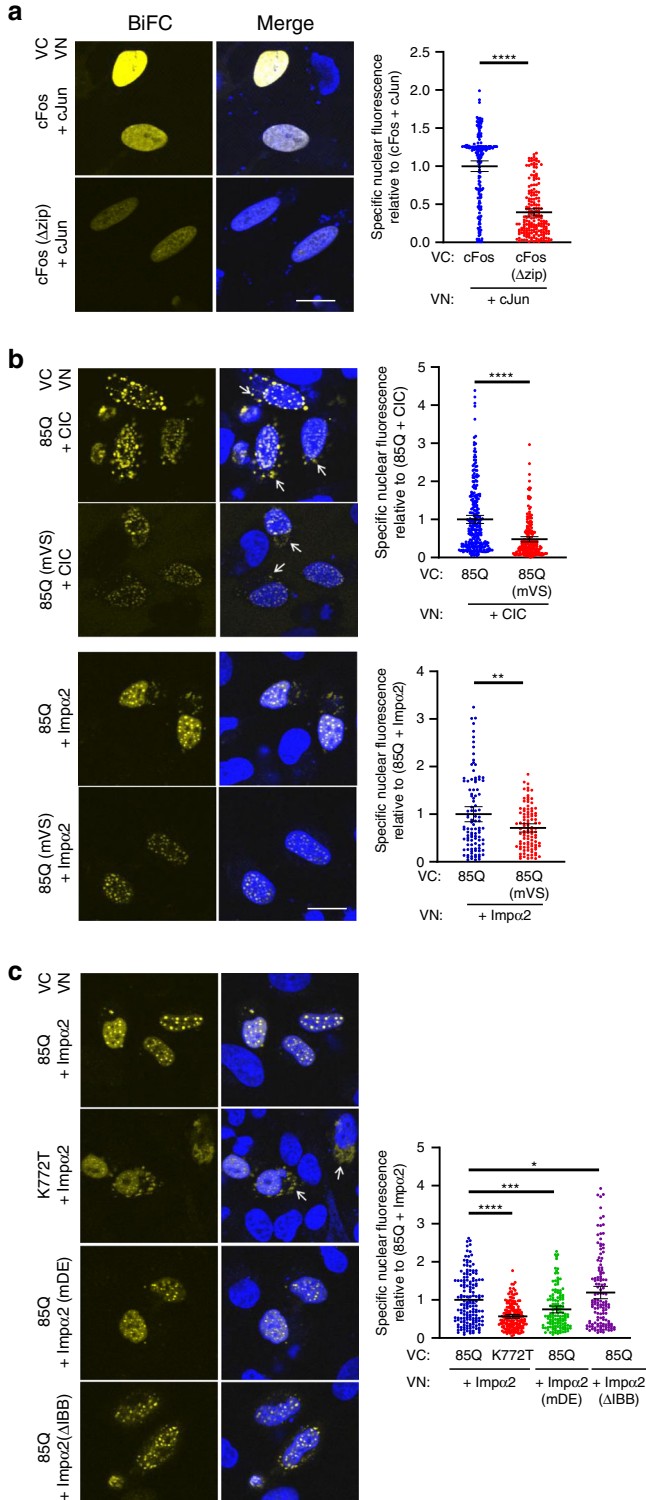

**Fig. 2 Ataxin-1[85Q] interacts with nuclear transporter protein importin-α2.** Cells were transfected to express the indicated BiFC constructs. At 18 h post-infection, intracellular protein-protein interactions were assessed by quantitative imaging of the complemented Venus fluorescence (BiFC, yellow, left panels), with merge overlays of BiFC and Hoechst nuclear stain (blue) (Merge, middle panels). Arrows indicate cytoplasmic complementation of the Venus fluorescent signal. Quantitative results (graphs, right panels) for specific nuclear fluorescence complementation of Venus represent the mean ± SEM for 100–200 cell nuclei per sample assessed across three independent experiments specifically as follows: **a** control interaction of VN-cJun complementing with VC-cFos (positive control) or VC-cFos (Δzip) (negative control) (n = 183 and 188 nuclei per group, respectively), p < 0.00001; **b** control interaction of VN-CIC complementing with wild-type ataxin-1[85Q] (positive control) or the mutant ataxin-1[85Q] mVS (negative control) as well as the test interaction of VN-Impα2 complementing with VC-wild-type ataxin-1[85Q] or VC-mVS mutant ataxin-1[85Q] (n = 258, 201, 101 and 101 nuclei per group, respectively), p < 0.00001 and p = 0.00295, respectively; **c** test interaction of wild-type VN-Impα2 with VC-wild-type ataxin-1[85Q] or VC-nuclear accumulation deficient ataxin-1[85Q] K772T mutant (n = 155 and 172 nuclei per group) or the test interaction of VN-Impα2 mDE or ΔIBB mutants with VC-wild-type ataxin-1[85Q] (n = 131 and 131 nuclei respectively), p value from top to bottom: p = 0.04089, p = 0.00042, p < 0.00001. **a–c** *p < 0.05; **p < 0.01; ***p < 0.001; ****p < 0.0001 (Two-sided Student's t test). Source data are provided as a Source Data file.

interaction with importin-α2 could contribute to the lowered toxicity seen for the ataxin-1 mVS mutant[42].

To probe further the biochemical nature of the ataxin-1[85Q]/ importin-α2 interaction, we examined additional approaches to disrupt the importin-recognition of cargo NLS. The ataxin-1 NLS mutant K772T, reported to disrupt ataxin-1 nuclear entry[28], was tested for its localisation, confirming distinctive cytoplasmic bodies formed by GFP-ataxin-1[85Q]K772T (Supplementary Fig. 5a). Notably, toxicity as assessed across the total population of GFP-protein expressing population was significantly lower for GFP-ataxin-1[85Q]K772T than for wild-type GFP-ataxin-1 (Supplementary Fig. 5b–d), emphasising the importance of relationships of ataxin-1 with nuclear transport and/or nuclear transporters. In the BiFC system, the ataxin-1[85Q]K772T + importin-α2 pair showed distinctive cytoplasmic bodies, but also with lower nuclear fluorescence (Fig. 2c). Whilst this may suggest a contribution to binding via an NLS-dependent mechanism, the different nuclear/cytoplasmic localisation of ataxin-1[85Q]K772T complicates interpretation. Thus, we chose to evaluate the mechanism(s) of interaction with two importin-α2 mutants, namely the importin-α2 NLS-binding deficient mutant[43] (importin-α2 D192K/E396R, abbreviated herein as importin-α2 mDE) and an importin-α2 mutant for which the importin-β-binding domain within its N-terminus was removed[44] (importin-α2 with residues 70-529 deleted, abbreviated herein as importin-α2 (ΔIBB)). Whereas nuclear fluorescence was significantly reduced for the ataxin-1[85Q] + importin-α2 D192K/E396R pair, a significant increase was noted for the ataxin-1[85Q] + importin-α2 ΔIBB pair. Thus, these analyses reinforce our proteomic identification of importin-α2 as an interacting partner for ataxin-1[85Q], with this interaction dependent on multiple domains of importin-α2 and at least partially dependent on the ataxin-1 NLS as shown by the importin-α2 mutants, but with some interaction remaining for the ataxin-1 NLS mutant. Collectively, these results reinforce a canonical relationship between a nuclear-localised protein such as ataxin-1 and nuclear transporters, and prompted us to question whether this relationship may be more complex with a potential impact of polyQ-ataxin-1 on nuclear transport processes as a mechanism contributing to SCA1.

system robustly reports the ataxin-1[85Q] interaction with CIC, dependent on the established interaction interface[42] including ataxin-1 residues V591 and S602.

In testing the ataxin-1[85Q] + importin-α2 pair, we again observed distinctive nuclear puncta; fluorescence was also partially but significantly decreased for the ataxin-1[85Q]mVS +importin-α2 pair (Fig. 2b). This latter observation suggests that ataxin-1[85Q]/importin-α2 interaction is also at least partially dependent on the interaction interface including ataxin-1 residues V591 and S602; this raises an interesting possibility that lowered

**PolyQ-ataxin-1 disrupts classical nuclear transport.** Previous studies have reported the ALS-causing C9orf72 repeat expansion RNA product to interact directly with the Ran regulator Ran GTPase activating protein 1 (RanGAP1), resulting in RanGAP1 mis-localisation and disruption of nucleocytoplasmic transport[8]. Significantly, RanGAP1, as well as the NPC component nucleoporin NUP62, have been described to co-localise with huntingtin protein aggregates[10].

To address whether polyQ-ataxin-1 may impact nuclear transport, we first examined well-characterised members of the importin (IMP) superfamily of nuclear transporters, importin-α2 (IMPα2), importin-β1 (IMPβ1) and exportin-1 (also known as CRM1). In classical nuclear protein import, IMPβ1 mediates the transfer of cargo proteins across the nuclear pore complex either directly or in conjunction with IMPα2 that acts as an adaptor binding both the cargo protein's NLS and IMPβ1[13]. When bound to RanGTP, CRM1 specifically recognises leucine-rich NESs to mediate protein export from the nucleus[45]. We co-expressed MBI-ataxin-1[85Q] with GFP-IMPα2, -IMPβ1, -CRM1, or GFP alone as a control to assess the effect of ataxin-1[85Q] on nucleocytoplasmic distribution under either control or arsenite stress conditions (Fig. 3). In the presence or absence of arsenite, GFP-IMPα2 was primarily nuclear, as previously observed[46], while GFP-IMPβ1 and -CRM1 were cytoplasmic and with a distinctive nuclear rim[47,48]; GFP was distributed throughout the cytoplasm and nucleus without a clear nuclear rim. In the presence of MBI-ataxin-1[85Q], however, GFP-IMPα2, -IMPβ1 and -CRM1 all showed prominent nuclear localisation with ataxin-1[85Q] that was further observed as striking nuclear bodies following arsenite stress exposure (Fig. 3a–c, denoted by arrowheads in Zoom panels); no pronounced nuclear body localisation was observed for the negative control of GFP alone (Fig. 3d). These observations of relocalisation to ataxin-1 nuclear bodies were further confirmed by quantitative evaluation of ataxin-1/GFP-IMPα2, ataxin-1/GFP-IMPβ1, and ataxin-1/GFP-CRM1 co-localisation as assessed by Pearson's correlation coefficient analysis; again, the negative control of GFP-only showed a very low Pearson's correlation coefficient indicating that GFP is not recruited into these nuclear bodies (Fig. 3e). We further confirmed that ataxin-1[30Q] expression similarly impacted GFP-IMPα2 and GFP-IMPβ1 localisation (Supplementary Fig. 6a,b,d), consistent with the notion that the elevated ataxin-1 levels permit nuclear body formation and influence nuclear transport protein distributions.

Mis-localisation of nuclear transporters could be expected to result in altered nucleocytoplasmic distribution of their cargo proteins, albeit that our observations that multiple importins as well as the exportin CRM1 are impacted in the presence of ataxin-1[85Q] (Fig. 3 and Supplementary Fig. 6) would complicate predictions of the magnitude or direction of the change to favour/disfavour nuclear import or export and the resulting change in nuclear to cytoplasmic ratio (Fn/c). Accordingly, we tested the impact of ataxin-1[85Q] or ataxin-1 [30Q] expression on the nuclear accumulation of a classical importin α/β1-recognised model cargo, GFP-NLS-βgal, containing the NLS from simian virus SV40 large tumour antigen (Fig. 4 and Supplementary Fig. 6c, respectively). We observed prominent nuclear localisation of GFP-NLS-βgal under both control and arsenite stress conditions, as expected, but we noted increased cytoplasmic levels of this protein under both conditions in the presence of ataxin-1 [85Q] or ataxin-1 [30Q] (Fig. 4a and Supplementary Fig. 6c, respectively, denoted by arrows). A mis-localisation of a portion of the GFP-NLS-βgal to the ataxin-1 nuclear bodies was also observed and confirmed by quantitative evaluation of ataxin-1/NLS-βgal co-localisation in the nucleus as assessed by Pearson's correlation coefficient analysis (Fig. 4b and Supplementary

Fig. 6d), consistent with an importin-cargo interaction that is not dissociated within the ataxin-1 nuclear bodies. The increased cytoplasmic levels of the cargo, GFP-NLS-βgal, were confirmed by image analysis of the GFP-NLS-βgal Fn/c in the presence of MBI-ataxin-1[85Q] (Fig. 4c and Supplementary Fig. 6e).

**PolyQ-ataxin-1 influences other nuclear import pathways.** To extend our survey of nuclear transporters in the presence of polyQ-ataxin-1 expression at the single-cell level, we next evaluated additional nuclear transport proteins. Comparable to our initial findings (Fig. 3), we observed mis-localisation of both importin-13 (IMP13)[49] (Fig. 5a), and Hikeshi, a non-importin that is responsible for nuclear import of heat shock protein 70 (HSP70) in response to heat shock[50] (Fig. 5b). Both GFP-IMP13 and GFP-Hikeshi were primarily nuclear in the presence or absence of arsenite stress, with partial localisation with ataxin-1 [85Q] under control conditions, and increased mis-localisation in the presence of arsenite-induced stress. These observations were further confirmed by quantitative evaluation of ataxin-1/GFP-IMP13 and ataxin-1/GFP-Hikeshi co-localisation as assessed by Pearson's correlation coefficient analysis (Fig. 5e). Consistent with these observations, we also observed alterations in the nucleocytoplasmic distribution for the IMP13 cargo, transcription factor NF-YB[51] (Fig. 5c; quantitative Fn/c shown in Fig. 5f), and the Hikeshi cargo, HSP70[50] (Fig. 5d; quantitative Fn/c shown in Fig. 5g), but also with mis-localisation of a portion of each GFP-cargo protein to the ataxin-1 nuclear bodies confirmed by quantitative evaluation of ataxin-1/NF-YB and ataxin-1/HSP70 co-localisation in the nucleus as assessed by Pearson's correlation coefficient analysis (Fig. 5e). Thus, in the presence of ataxin-1 [85Q], the changes in importins as well as CRM1 have the capacity to reduce the nuclear accumulation of some proteins (as exemplified by GFP-NLSβ-gal, Fig. 4) whilst increasing nuclear accumulation of other proteins (as exemplified by NF-YB and HSP70, Fig. 5).

In contrast to the results above, we observed that a number of other nuclear transporters proteins were not mis-localised by polyQ-ataxin-1 expression. Specifically, GFP-IMPα4, -IMP7, and -IMP16 did not show substantial recruitment to the polyQ-ataxin-1 nuclear bodies under control or arsenite stress conditions (Supplementary Fig. 7a–c). GFP-IPO5 (IMP5), in contrast, appeared to localise to the polyQ-ataxin-1 nuclear structures, but only in the absence of arsenite-induced stress (Supplementary Fig. 7d, denoted by arrowheads in Zoom panel). These observations were further confirmed by quantitative evaluation of ataxin-1/transporter co-localisation as assessed by Pearson's correlation coefficient analysis (Supplementary Fig. 7e). Taken together, these observations highlight selectivity in the impact of the polyQ-ataxin-1 nuclear bodies by disrupting multiple, but by not all, nuclear import pathways.

The nuclear protein transport pathways mediated by importins/exportins are Ran-dependent and so we also assessed any impact of polyQ-ataxin-1 on RanGTP/GDP distribution through altering localisation of its regulators, the Ran GTPase activating protein RanGAP1, and the Ran guanine nucleotide exchanger RCC1. RanGAP1 can localise with mutant huntingtin or C9orf72 polyGA aggregates in in vivo models of HD and ALS[8,9], but we observed no alteration in RanGAP1 subcellular distribution or co-localisation with ataxin-1 nuclear bodies in the presence of polyQ-ataxin-1 expression (Supplementary Fig. 8). With our proteomics data identifying RanGAP1, as well as Ran-binding protein 1 (RanBP1), in association with GFP-ataxin-1[85Q] in our pulldown analysis (Supplementary Data 1), the implication is that significant mis-localisation of RanGAP1 requires longer-term exposure to polyQ-ataxin-1 aggregates, as observed for huntingtin-RanGAP1

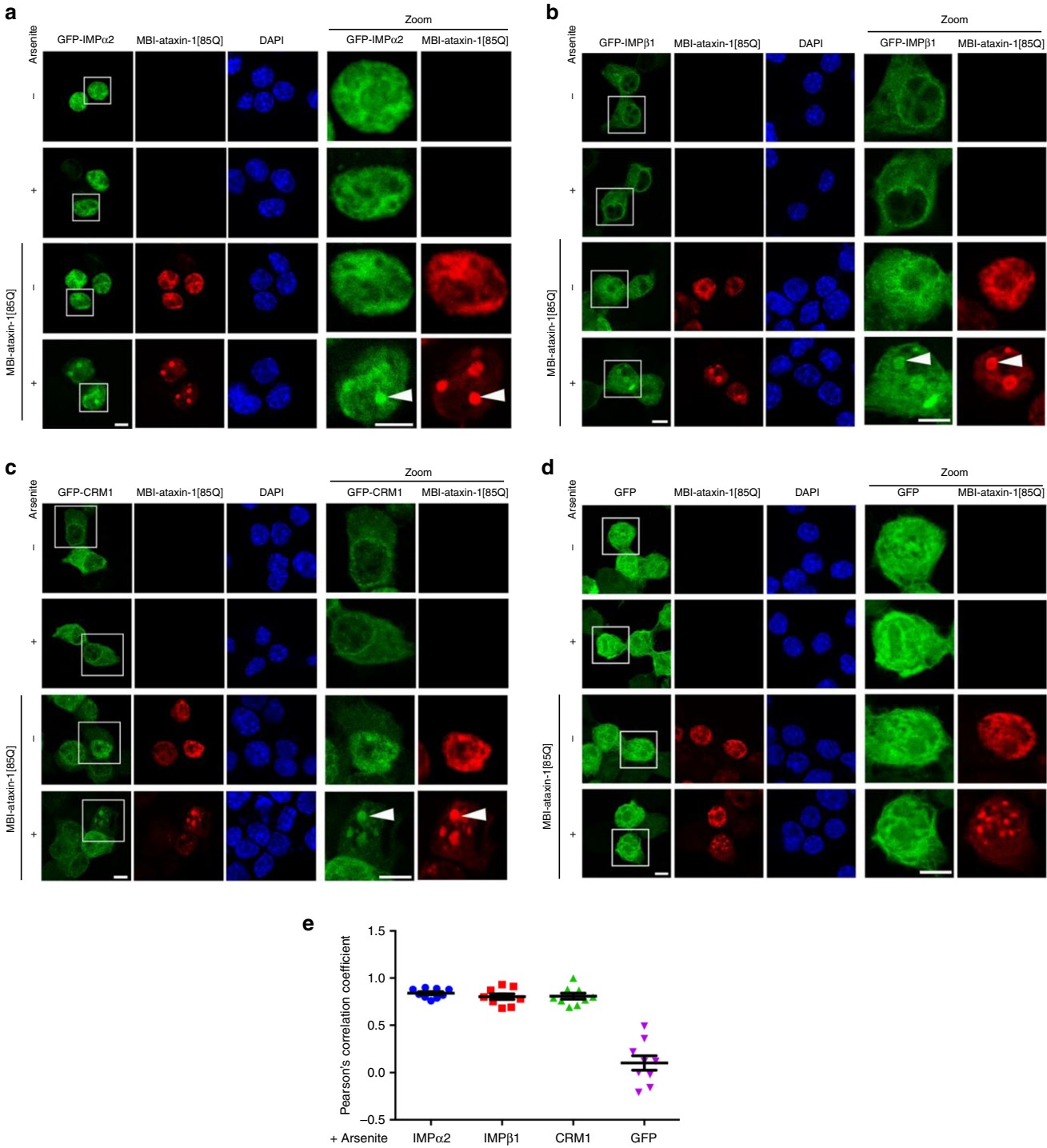

**Fig. 3 Ataxin-1[85Q] induces mislocalisation of classical nuclear transport proteins.** Neuro-2a cells were cotransfected to express MBI-ataxin-1[85Q] together with (**a**) GFP-Importin-α2 (IMPα2), (**b**) GFP-Importin-β1 (IMPβ1), (**c**) GFP-CRM1, or (**d**) GFP alone. At 24 h post-transfection, cells were treated with arsenite as indicated, and then fixed, stained with anti-myc antibodies and DAPI, before CLSM imaging. Representative images of cells are shown, with zoom images (right panels) corresponding to the boxed regions. Mis-localisation to the ataxin-1 nuclear bodies is indicated by the white arrowheads. Scale bar = 10 μm. **e** Pearson's correlation coefficients, calculated using the coloc2 plugin for localisation between fluorophores in a 25 μm$^2$ square centred over each arsenite-induced ataxin-1 nuclear body, quantitatively assessed co-localisation of GFP-tagged proteins with ataxin-1 nuclear bodies, where 0 indicates no correlation between two channels whereas 1 indicates complete correlation of two channels. Results represent the mean ± SEM (n = 9 cells imaged across three independent experiments). Source data are provided as a Source Data file.

association in vivo[9]. In contrast, our assessment of endogenous RCC1 detected with an anti-RCC1 antibody or ectopically addressed GFP-RCC1 revealed localisation with polyQ-ataxin-1 nuclear bodies (Supplementary Fig. 9). Thus, through mis-localising RCC1 into distinct nuclear bodies, polyQ-ataxin-1 has

the potential to reduce nuclear RanGTP levels, and hence importin/exportin-dependent nuclear trafficking. Additionally, not all RCC1 is sequestered into polyQ-ataxin-1 nuclear bodies, explaining how there is not complete disruption of the localisation of all nuclear transporters/nuclear transport pathways.

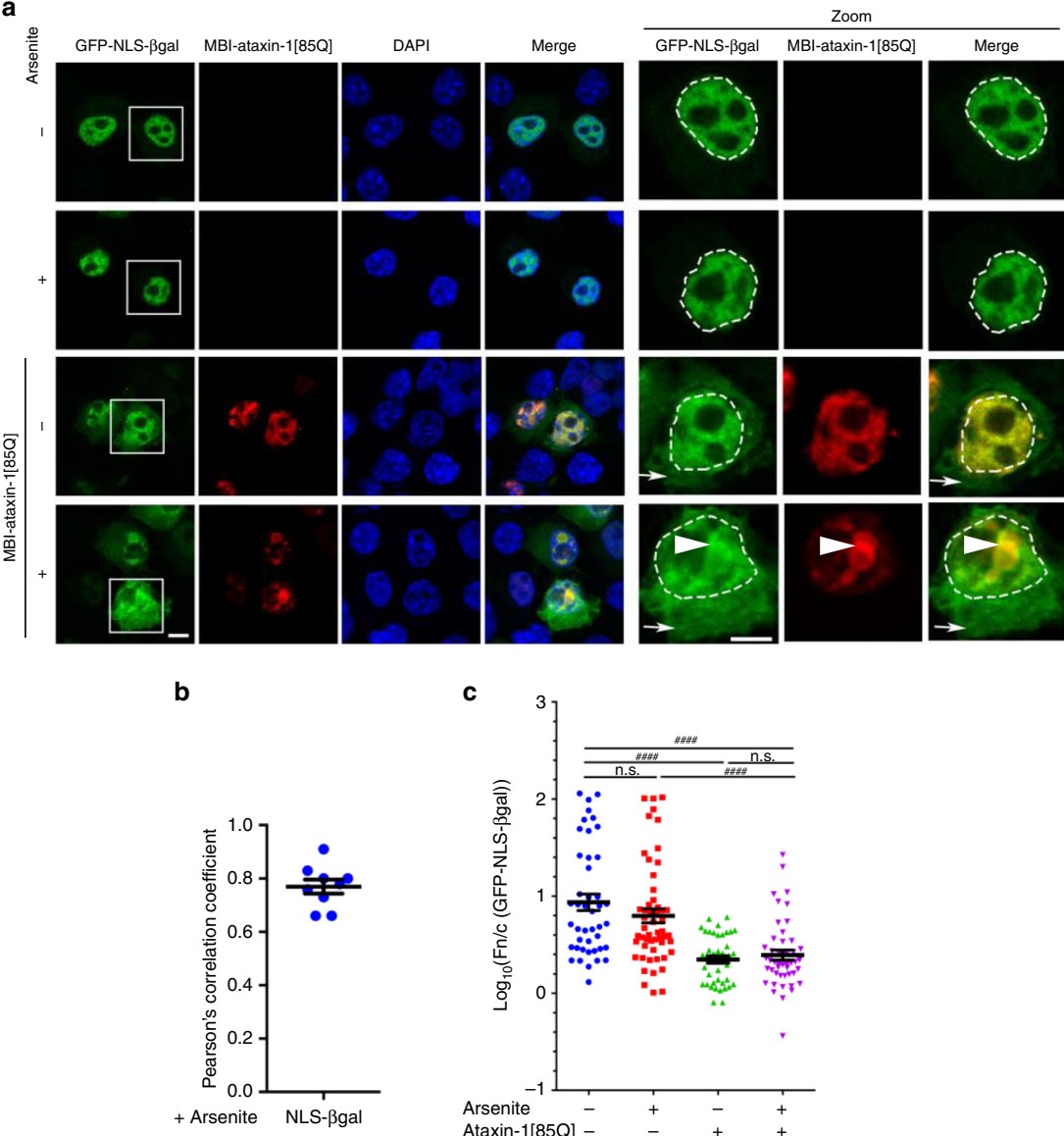

**Fig. 4 Ataxin-1[85Q] reduces nuclear accumulation of a conventional importin-α/β1-cargo.** Neuro-2a cells were cotransfected to express MBI-ataxin-1 [85Q] together with GFP-TagNLS-β-galactosidase (GFP-NLS-βGal). At 24 h post-transfection, cells were treated with arsenite as indicated, and then fixed, stained with anti-myc antibody and DAPI, before CLSM imaging. **a** Representative images are shown; merge panels overlay GFP, myc, and DAPI images. Zoom images (right panels) correspond to the boxed regions. The position of the nucleus, as determined by DAPI staining, is indicated by the white dashed lines; white arrows denote increased cytoplasmic fluorescence in the presence of MBI-ataxin-1[85Q]. Scale bar = 10 μm. **b** Pearson's correlation coefficients, calculated using the coloc2 plugin for localisation between fluorophores in a 25 μm$^2$ square centred over each arsenite-induced ataxin-1 nuclear body, quantitatively assessed co-localisation of GFP-NLS-βgal with ataxin-1 nuclear bodies, where 0 indicates no correlation between two channels whereas 1 indicates complete correlation of two channels. Results represent the mean ± SEM ($n = 9$ cells imaged across three independent experiments). **c** Integrated fluorescence intensity in nucleus and cytoplasm was estimated and the nuclear to cytoplasmic fluorescence ratio (Fn/c) determined using a modified CellProfiler pipeline. Results represent the mean log$_{10}$ Fn/c ± SEM calculated across three independent experiments using the following numbers of analysed cells: -arsenite/-ataxin-1[85Q] $n = 44$, +arsenite/-ataxin-1[85Q] $n = 53$, -arsenite/+ataxin-1[85Q] = 43, +arsenite/+ataxin-1[85Q] $n = 46$, $p$ value from top to bottom and left to right: $p < 0.00001$, $p < 0.00001$, $p > 0.99999$, $p > 0.99999$, $p = 0.00005$ (####$p < 0.0001$, n.s. not significant, Mann–Whitney and Kruskal–Wallis non-parametric test). Source data are provided as a Source Data file.

**PolyQ-ataxin-1 impacts importin localisation in vivo**. With these observations of acute disruptions of nuclear transport when polyQ-ataxin-1 is expressed in cultured cells, we next addressed the disruption of nuclear transport in a SCA1 mouse model. We thus extended our in vitro studies to assess the impact of longer term expression of polyQ-ataxin-1 in vivo, and in particular if disruption of nuclear transport events would be observed in the Purkinje (calbindin-expressing) cells of a SCA1 mouse model. Importantly, our in vivo studies could be performed by analysing

the localisation of endogenously expressed proteins, and so can provide an assessment independent of epitope-tag position or identity. Therefore, we stained brain sections from wild-type (FVB) and *ATXN1*[82Q] transgenic mice for importin-β1, ataxin-1 and calbindin, and further included two transgenic lines as controls (Fig. 6). One line was *ATXN1*[30Q]S776D (i.e. expressing the wild-type *ATXN1*[30Q] but with the phosphomimetic mutation S776D that induces SCA1 disease initiation and Purkinje cell dysfunction but without late-stage Purkinje cell

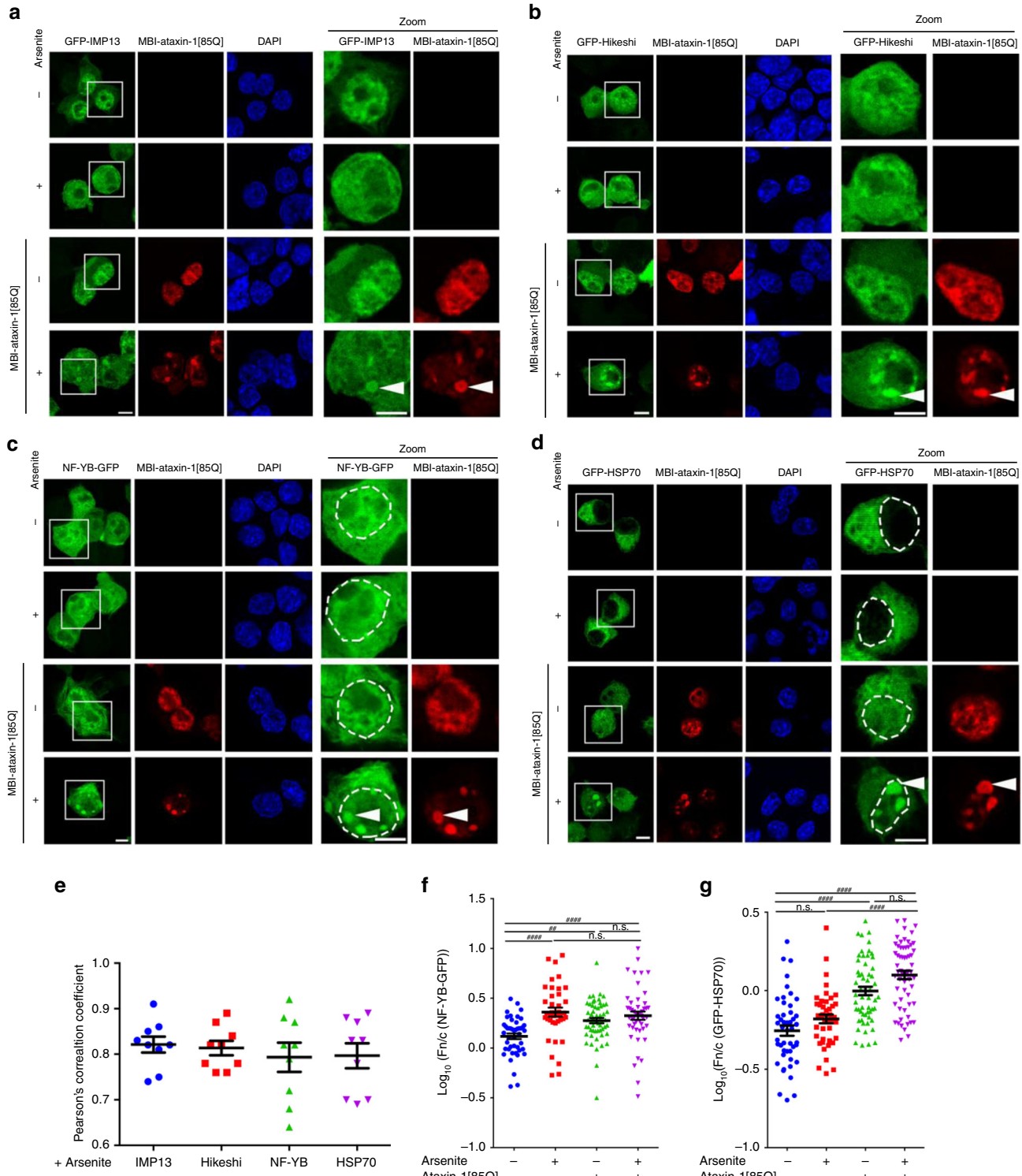

death[23]). The second line was *ATXN1* [82Q]CIC(mVS), expressing the CIC-interaction-deficient *ATXN1*[82Q] mutant V591A/S602D that we also demonstrated to have reduced importin-α2 binding (Fig. 2b), and that does not develop SCA1 disease pathology[42]. For the analyses of the control FVB mice, we noted prominent nuclear boundary staining for importin-β1 (Fig. 6a,b, upper panels); this reinforced our observations of the nuclear rim of GFP-importin-β1 detected in our control cultured cells and as reported elsewhere[54], but also suggested that cytoplasmic GFP-importin-β1 in those cultured cells (Fig. 3b upper panels) may

result from higher levels of expressed importin-β1. For the *ATXN1*[82Q] animals, the nuclear rim of importin-β1 was disrupted (Fig. 6a,b), with quantitative analyses indicating a significantly increased Purkinje cell population with disruption scored as moderate-severe (Fig. 6c). Further analysis of a noted prominent nuclear rim localisation further confirmed no disruption in the Purkinje cells analysed from *ATXN1*[30Q]S776D and *ATXN1*[82Q]CIC(mVS) animals (Fig. 6c). Interestingly, these observations reinforce the links between end-stage SCA1 pathology and altered localisation of nuclear transport proteins as

**Fig. 5 Ataxin-1[85Q] alters localisation and transport activity of importin-13 and hikeshi.** Neuro-2a cells were transfected to co-express MBI-ataxin-1 [85Q] with (**a**) GFP-Importin-13 (GFP-IMP13), (**b**) GFP-Hikeshi, (**c**) the importin-13 cargo protein NF-YB-GFP or (**d**) the hikeshi cargo protein GFP-HSP70. At 24 h post-transfection, cells were treated with arsenite as indicated, and then fixed, stained with anti-myc antibody and DAPI, before CLSM imaging. Representative images are shown; merge panels overlay GFP, myc, and DAPI images. Zoom images (right panels) correspond to the boxed regions; mislocalisation is denoted by the white arrowheads whereas in **c**, **d** the position of the nucleus, as determined by DAPI staining, is indicated by the white dashed lines. Scale bar = 10 μm. **e** Pearson's correlation coefficients, calculated using the coloc2 plugin for localisation between fluorophores in a 25 μm$^2$ square centred over each arsenite-induced ataxin-1 nuclear body, quantitatively assessed co-localisation of GFP-tagged proteins with ataxin-1 nuclear bodies, where 0 indicates no correlation between two channels whereas 1 indicates complete correlation of two channels. Results represent the mean ± SEM ($n = 9$ cells imaged across three independent experiments). **f**, **g** Integrated fluorescence intensity in nucleus and cytoplasm was estimated and the nuclear to cytoplasmic fluorescence ratio (Fn/c) determined using a modified CellProfiler pipeline. Results represent the mean $\log_{10}$ Fn/c ± SEM calculated across three independent experiments using the following numbers of analysed cells: −arsenite/−ataxin-1[85Q] $n = 46$ (**f**) or 47 (**g**), +arsenite/−ataxin-1 [85Q] $n = 39$ (**f**) or 43 (**g**), −arsenite/+ataxin-1[85Q] = 56 (**f**) or 57 (**g**), +arsenite/+ataxin-1[85Q] $n = 48$ (**f**) or 60 (**g**), $p$ value from top to bottom and left to right: **f** $p = 0.00003$, $p = 0.00139$, $p > 0.99999$, $p < 0.00001$, $p > 0.99999$, **g** $p < 0.00001$, $p < 0.00001$, $p = 0.21628$, $p > 0.99999$, $p < 0.00001$ (*$p < 0.05$, **$p < 0.01$, ****$p < 0.0001$, n.s. not significant, Mann–Whitney and Kruskal–Wallis non-parametric test). Source data are provided as a Source Data file.

exemplified by importin-β1, i.e. the disruption of nuclear transport proteins in the in vivo setting appears to be more closely aligned with the progressive disease to late-stage dysfunction observed in *ATXN1*[82Q] animals than the features of early dysfunction that do not proceed to Purkinje cell loss in the *ATXN1*[30Q]S776D animals.

**PolyQ-ataxin-1 impacts the nuclear pore complex in vivo.** The nuclear pore complex (NPC) is the regulated gateway for trafficking between the cell cytosol and nucleus, and is a potential target of mutant polyQ-huntingtin protein in HD[10], where progressive disruption of nuclear envelope morphology has been observed in aging mice expressing the polyQ-huntingtin protein[9]. Post-mortem samples from patients with ALS[52] also show disruptions to the nuclear envelope, whereas polyQ-ataxin-1 may also decrease nuclear membrane instability[53]. Among the nucleoporins, the main components of NPC, NUP98 has the most conserved FG-dipeptide motif domain and is a major contributor to the NPC permeability barrier[54]. Indeed, the more cohesive nature of NUP98 GLFG repeat domains compared to the FxFG repeats of other nucleoporins[55] can drive the formation of highly concentrated gels[56]. Furthermore, and of relevance to the studies of neurodegenerative diseases, NUP62 was reported to co-localise with huntingtin protein aggregates in a HD mouse model[10]. We therefore set out to evaluate the effect of polyQ-ataxin-1 on nucleoporins NUP98 and NUP62 to evaluate any possible disruption of NUP distribution in the Purkinje cells in SCA1 mice.

We stained brain sections from wild-type (FVB) and *ATXN1* [82Q] transgenic mice for NUP98 or NUP62 along with ataxin-1 and calbindin; we again included the two transgenic lines *ATXN1* [30Q]S776D[23], and *ATXN1*[82Q]CIC(mVS) as controls (Fig. 7). We noted disruption to the nuclear rim morphologies as detected by staining for either NUP98 or NUP62, but without noticeable relocalisation to the ataxin-1 nuclear bodies (Fig. 7a,c). Further quantitative analysis (50–100 cells/animal, 200–350 cells total) by an observer blinded to the animal genotype, and defining categories of either normal-mild (nuclear rim staining round to mostly round with <2 small to medium sized wrinkles) or moderate-severe (nuclear rim staining observed to be more elongated with large invaginations and distorted morphology) confirmed disruption in the *ATXN1*[82Q] but not *ATXN1*[30Q] S776D and *ATXN1*[82Q]CIC(mVS) animals (Fig. 7b,d). Thus, the disruption of NUP98 and NUP62 staining patterns only in the *ATXN1*[82Q] transgenic mice leads to the suggestion that nuclear pore changes are of highest relevance in progressive SCA1 pathology but likely to be less critical in the early dysfunction seen in the *ATXN1*[30Q]S776D model in vivo despite our

observations that acute overexpression of ataxin-1[30Q] or ataxin-1[85Q] in vitro can initiate comparable toxicity and changes in nuclear transport. Thus, the translation of findings from the scenario of in vitro cell biology to in vivo testing in animal models is critically important in understanding the mechanisms contributing to long term pathology. Previously, altered NUP62 localisation has been observed in the presence of mutant huntingtin protein in HD[10] and altered NUP98 localisation in the presence of tau in AD[11] highlighting the potential for a shared impact on nuclear transport in these neurodegenerative diseases.

**Ataxin-1 potentially impacts multiple other pathways.** Previous genetic screens have highlighted the impact of nuclear transport proteins across a number of neurodegenerative diseases. In exploring suppressors of C9orf72 repeat expansion toxicity in a *Drosophila* ALS model, multiple NPC proteins, importins/exportins and Ran regulators have been identified[7,8]. The earlier screens for genetic suppressors of ataxin-1-induced neuronal degeneration in a *Drosophila* SCA1 model previously identified the nucleoporin NUP44A as a suppressor of toxicity[34]. Our study investigates the impact of short-term polyQ-ataxin-1 expression on cellular nuclear transport processes, the findings indicating that nuclear transport is directly disrupted by short-term polyQ-ataxin-1 expression. Our proteomics analyses, in combination with our surveying of additional nuclear transporters, Ran regulators and nucleoporins, point to the disruption at multiple critical points across the cellular nuclear transport system. Notably, environmental stress induced by arsenite, or alternatively via direct application of $H_2O_2$ as a pro-oxidant stress (Supplementary Fig. 10), can exacerbate these changes.

Our results emphasise that approaches to rescue the nuclear transport disruption by mutant ataxin-1 may ultimately give insight into possible therapeutic interventions to slow neurodegenerative disease progression in SCA1. Furthermore, the ataxin-1 interactome remains a resource for further exploration. For example, we identified replication protein A1 (Rpa1) by pulldown approaches (Supplementary Data 1); Rpa1 has been previously identified as having the largest effect on lifespan in the *Drosophila* SCA1 model and has been located at the hub position linked to repair systems such as homologous recombination[57]. Others have implicated genes in the protein folding/heat-shock response or ubiquitin-proteolytic pathways as modifiers of toxicity in models of SCA1[34] and our proteomics studies have identified multiple heat shock proteins as well as proteins involved in ubiquitination, and ubiquitin itself, as part of the ataxin-1 interactome (Supplementary Data 1). Ataxin-1 phosphorylation may also influence ataxin-1 stability[58]; with the implication of cAMP-

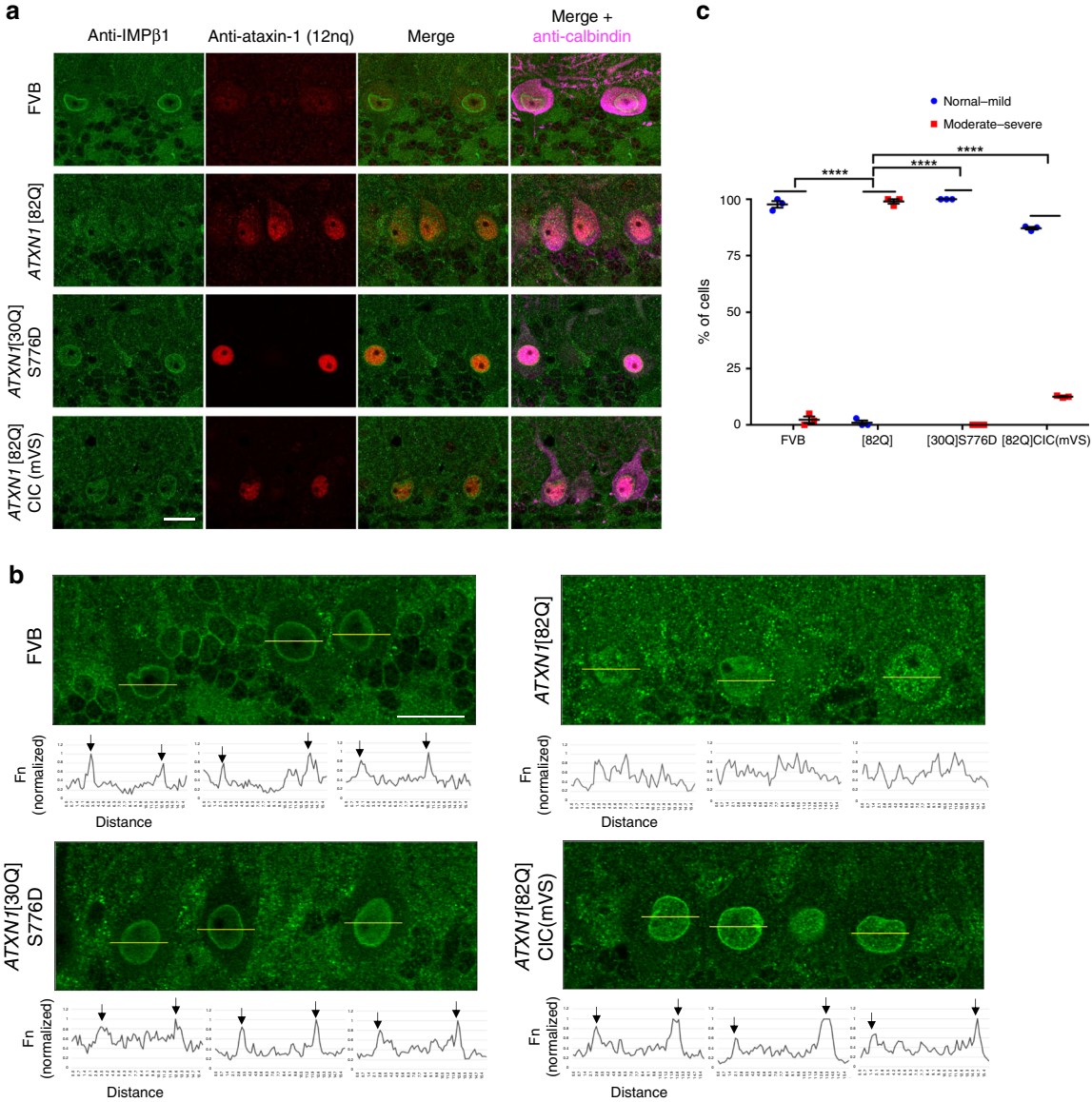

**Fig. 6 Importin-β1 is mislocalised in Purkinje cells in SCA1 mice.** Mice (12 weeks old) of each genotype (FVB, *ATXN1*[82Q], *ATXN1*[30Q]S776D and *ATXN1*[82Q]CIC(mVS)) were anaesthetised and cerebellum sections cut for staining and imaging. **a** IMPβ1 (green), ataxin-1 (red) and calbindin (magenta, a marker of Purkinje cells) were detected by immunofluorescent staining. Merge panels (IMPβ1 + ataxin-1 or IMPβ1 + ataxin-1+calbindin) are shown. **b** Quantitative analysis of Purkinje cell IMPβ1 fluorescence intensity in from each genotype; regions of the fluorescence measurements are indicated by the yellow lines (upper panels) and fluorescence intensity peaks are indicated by black arrows (lower panels). **a, b** Scale bars = 20 μm. **c** IMPβ1 staining was scored as normal-mild (strong nuclear rim staining indicated by 2-3 fluorescence peaks corresponding to nuclear rim by line scan measurement by Fiji), or moderate-severe (weak nuclear rim staining indicated by irregular fluorescence peak distribution across the nucleus) by an observer blinded to the animal genotype; results were calculated as the percentage of Purkinje cells of each type. Results represent the mean ± SEM calculated using the following numbers of analysed cells in mice: FVB $n = 161$ cells from 3 mice, [82Q] $n = 171$ cells from 3 mice, [30Q]S776D $n = 179$ cells from 3 mice, [82Q] CIC(mVS) $n = 106$ cells from 3 mice, $p$ value (both "normal – mild" and "moderate – severe") FVB vs [82Q] < 0.00001, [82Q] vs [30Q]S776D < 0.00001, [82Q] vs [82Q]CIC(mVS) < 0.00001 (****$p < 0.0001$, two-way ANOVA and Tukey's multiple comparisons test). Source data are provided as a Source Data file.

dependent protein kinase as a mediator of phosphorylation and toxicity of ataxin-1[22], it is of interest that we identified both regulatory and catalytic subunits of cAMP-dependent protein kinase in our interactome analysis (Supplementary Data 1). Increasing levels of wild-type ataxin-1 are sufficient to alter certain features of agility[37] and have been shown in some models to lead to SCA1-like neurodegeneration[34–36], and thus the exact balance of binding partner selection by ataxin-1 appears to be of critical importance. With further details of binding partner selection that will undoubtedly be revealed by proximity labelling

in combination with quantitative proteomics approaches[59] including higher-order multiplexing techniques[60], we propose that deciphering the cellular events underlying the ataxin-1 interactome may well be the key to future treatment options for SCA1 and other neurodegenerative diseases.

## Methods
**Plasmids**. The GFP-ataxin-1[85Q] and GFP-ataxin-1[30Q] plasmids were provided by D.M. Hatters (University of Melbourne). Myc-BioID (MBI)-ataxin-1 [85Q], the myc-tagged BirA* N-terminal fusion with ataxin-1[85Q] for use in the

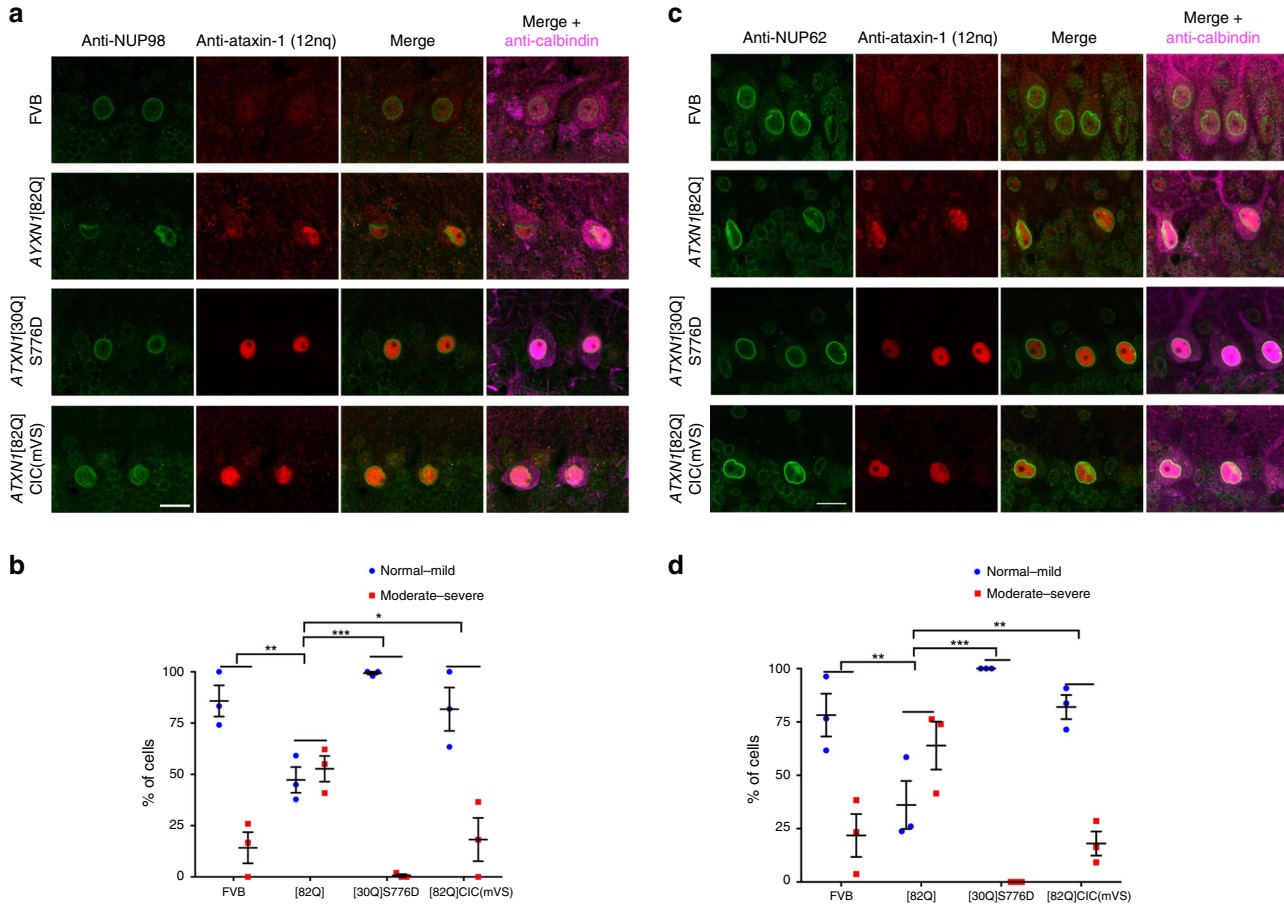

**Fig. 7 Nucleoporins NUP98 and NUP62 are mislocalised in Purkinje cells in SCA1 mice.** Mice (12 weeks old) of each genotype ((FVB, *ATXN1*[82Q], *ATXN1*[30Q]S776D and *ATXN1*[82Q]CIC(mVS)) were anaesthetised and cerebellum sections were cut for staining and imaging. NUP98 (green), ataxin-1 (red) and calbindin (magenta, a marker of Purkinje cells) were detected by immunofluorescent staining. Merge panels **a** NUP98 + ataxin-1 or NUP98 + ataxin-1+calbindin and **c** NUP62 + ataxin-1 or NUP62 + ataxin-1+calbindin are shown. **a**, **c** Scale bars = 20 μm. **b** NUP98 nuclear rim staining or **d** NUP62 snuclear rim staining was scored as normal-mild (round to mostly round with ≤2 small to medium-sized wrinkles) or moderate-severe (elongated with large invaginations and distorted morphology) by an observer blinded to the animal genotype; results were calculated as the percentage of Purkinje cells of each type. Results represent the mean ± SEM calculated using the following numbers of analysed cells in mice: FVB $n = 343$ (**b**) or 365 (**d**) cells from 3 mice, [82Q] $n = 291$ (**b**) or 347 (**d**) cells from 3 mice, [30Q]S776D $n = 290$ (**b**) or 271 (**d**) cells from 3 mice, [82Q]CIC(mVS) $n = 228$ (**b**) or 182 (**d**) cells from 3 mice, $p$ value (both "normal – mild" and "moderate – severe") (**b**) FVB vs [82Q] = 0.00818, [82Q] vs [30Q]S776D = 0.00056, [82Q] vs [82Q]CIC(mVS) = 0.01818, (**d**) FVB vs [82Q] = 0.00939, [82Q] vs [30Q]S776D = 0.00020, [82Q] vs [82Q]CIC(mVS) = 0.00477 ($*p < 0.05$, $**p < 0.01$, $***p < 0.001$, two-way ANOVA and Tukey's multiple comparisons test). Source data are provided as a Source Data file.

BioID system, was generated by amplifying the coding region of ataxin-1[85Q] from the GFP-ataxin-1[85Q] construct to include the desired restriction enzyme sites (EcoRI and HindIII), and then ligation into the pcDNA3.1_mycBioID plasmid[27] (#35700, addgene). The PCR primer pairs used were 5'-GCGAATTCAT GAAATCCAACCAAGAGCGGAGC-3' (forward) and 5'-GCAAGCTTCTACTTG CCTACATTAGACCGGCC-3' (reverse); this strategy was also used to generate the MBI-ataxin-1[30Q] plasmid as a control. VC-ataxin-1[85Q], the Venus fluorescent protein C-terminal fragment (Venus amino acids 155-238, A206K) fused with ataxin-1[85Q] for use in the BiFC system, was generated from PCR amplification using the primers 5'-GGCCAAGAATTCGGAAATCCAACCAAG-3' and 5'-CCGGTTGGTACCCTTGCCTACATTAGACC-3' and inserted into the EcoRI/ KpnI restrictions sites of plasmid vector pBiFC-VC155 plasmid[38] (#22011, addgene). VC-ataxin-1[85Q]mVS, encoding the Venus fluorescent protein C-terminal fragment fused with a mutant of ataxin-1 (ataxin-1[85Q] V591A S602D) with lowered CIC-interaction[42] as well as VC-ataxin-1(K772T), encoding the Venus fluorescent protein C-terminal fragment fused with a mutant of ataxin-1 (ataxin-1[85Q] K772T) with lowered nuclear localisation[28] were created using the Gibson Assembly® Cloning Kit (New England Biolabs). The first strand was created using the forward primer:
5'-CAGATCTCGAGCTCAAGCTTCGAATTCGAAATCCAACCAAGAGC GGAGC-3' with the reverse primer 5'-GAAATCTTCTGTTTTTAAGTCTT CCGCCTTCTTTAGC-3' (for the ataxin-1[85Q]mVS mutant); and with the reverse primer 5'- CGACCACCTCCTCGTCCTCGTTGCC-3' (for the ataxin-1 [85Q]K772T mutant). The second strand was created using the forward primer 5'-GTGGATCCCGGGCCCGCGGTACCCTACTTGCCTACATTAGACC-3' and

with the reverse primer 5'- GACTTAAAAACAGAAGATTTCATCCAGGATGCA GAG-3' (for the ataxin-1[85Q]mVS mutant.); and with the reverse primer 5'- GG CAACGAGGACGAGGAGGTGGTCG-3' (for the ataxin-1[85Q]K772T mutant). The construct was created by annealing both strands into EcoRI/KpnI restriction sites of pEGFP-C1. The mutant constructs were then PCR-amplified using the primers 5'- GGCCAAGAATTCGGAAATCCAACCAAG-3' and 5'- CCGGTTG GTACCCTTGCCTACATTAGACC-3' and inserted into the EcoRI/KpnI restrictions sites of the pBiFC-VC155 plasmid[38] (#22011, addgene). All resulting constructs were validated by restriction digestion and full sequence analysis.

To assess ataxin-1[85Q] interactions with identified binding partners using the BiFC system[38], we generated multiple plasmids using the Venus fluorescent protein N-terminal fragment (Venus amino acids 1-154, I152L) fused with the binding partners CIC (VN-CIC) and wild-type murine importin-α2 (VN-Impα2) using the pBiFC-VN155(I152L) plasmid[38] (#27097, addgene). In addition, importin-α2 mutants, Imp-α2 mDE (Imp-α2 D192K E396R) mutated to reduce interactions with classical nuclear localisation sequences[43], and Imp-α2 ΔIBB ((Imp-α2 70-529) to remove the importin-β-binding domain[44] were used. Imp-α2 mDE was obtained from Y. Yoneda (National Institutes of Biomedical Innovation, Health and Nutrition (NIBIOHN), Japan). Imp-α2 ΔIBB was generated from PCR amplification using the primers 5'-GGCCAAGAATTCGGAACCAGGGTACTG TAAATTGG-3' and 5'-CCGGTTGGTACCGAAGTTAAAGGTCCCAGGAGC-3' and inserted into EcoRI/KpnI restriction sites of plasmid vector pBiFC-VN155 (I152L) (#27097 addgene). Importin-α2 and importin-α2 mDE were PCR-amplified using the primers 5'- GGCCAAGAATTCGGTCCACGAACGAGAATG-3' and 5'-CCGGTTGGTACCGAAGTTAAAGGTCCCAGGAGC-3' and inserted

into the EcoRI/KpnI restriction sites of pBiFC-VN155(I152L) (#27097 addgene). All primers are also listed in Supplementary Table 3.

All resulting constructs were validated by restriction digestion and full sequence analysis. As controls in the BiFC assays, we assessed Fos/c-Jun interactions using VC-cFos (#22013, addgene), VC-cFos(Δzip) (#22014 addgene) and VN-cJun (#27098 addgene).

In addition, to assess the influence of ataxin-1 on nuclear transport in Neuro-2a cells, we assessed a range of nuclear proteins, choosing GFP-tagged proteins to ensure consistent comparisons when co-transfected with MBI-ataxin-1: GFP or GFP-ataxin-1[85Q]S776A (generated by site-directed mutagenesis of the GFP-ataxin-1[85Q] construct); GFP-IMPα2 and GFP-IMPα4[61]; GFP-IMPβ1[44]; GFP-CRM1 and GFP-IPO5 (generated by Gateway cloning); GFP-NLS-βgal[62]; GFP-IMP13[63]; GFP-Hikeshi, GFP-Hsp70[64]; NF-YB-GFP[65]; GFP-IMP7 (provided by R Seger, Department of Biological Regulation, The Weizmann Institute of Science, Rehovot, Israel), GFP-IMP16 (also known as RanBP16 or exportin-7)[66]; GFP-RCC1 (provided by Y Miyamoto, National Institutes of Biomedical Innovation, Health and Nutrition, Osaka, Japan).

**Cell culture, transfection and treatment**. Mouse neuroblastoma cells (Neuro-2a) (ATCC® CCL-131™, original line provided by D.M. Hatters, University of Melbourne) were used in a majority of experiments and were cultured in Opti-MEM supplemented with 2 mM L-glutamine (Gibco), 10% (v/v) foetal bovine serum (BOVOGEN) and 100 U/ml penicillin/streptomycin (Gibco). Cells were plated and cultured (16 h), transfected with the indicated constructs using Lipofectamine 2000 reagent (Invitrogen) according to the manufacturer's instructions (24 h), and then exposed as indicated to 300 μM sodium arsenite (Sigma) or hydrogen peroxide (H₂O₂) (ChemSupply) for 1 h. GFP was included as a control for imaging and proteomics protocols examining GFP-ataxin-1[85Q]. When BioID protocols were used, biotin (50 μM) was included during the transfection protocols with the MBI vector (for blot detection) or MBI-ataxin-1[85Q] (for blot detection or proteomics protocols). In control experiments, Ad293 (Strategene, now Agilent, 240085) or Hela cells (ATCC® CCL-2™) were used. Ad293 cells were cultured in Dulbecco's Modified Eagle's media (DMEM) supplemented with 10% (v/v) foetal bovine serum (FBS), 100 U/ml penicillin/streptomycin (Gibco); cells were transfected using Lipofectamine 2000 reagent (Invitrogen) and grown for 24 h before analysis. Hela cells were maintained in DMEM supplemented with 10% FBS, 1 x GlutaMAX (Thermofisher, USA); cells were transfected using Lipofectamine 3000 (Invitrogen) and grown for 18 h before analysis.

**Cell lysate preparation and pull-down protocols**. Cell lysates were prepared using RIPA buffer [50 mM Tris-HCl, pH 7.3, 150 mM NaCl, 0.1 mM ethylene-diaminetetraacetic acid (EDTA), 1% (w/v) sodium deoxycholate, 1% (v/v) Triton X-100, 0.2% (w/v) NaF and 100 μM Na₃VO₄] supplemented with complete protease inhibitor mix (Roche Diagnostic). Cell lysates were incubated on ice (20 min) and cleared by centrifugation (940 g, 20 min). Protein concentrations were determined using the BioRad assay.

For BioID pull-down prior to mass spectrometry or immunoblot analysis, whole-cell lysates (3 mg protein) prepared from Neuro-2a cells expressing MBI-ataxin-1[85Q] were incubated with streptavidin-agarose (0.05 mL) (Invitrogen). For GFP pull-down prior to mass spectrometry or immunoblot analysis, whole-cell lysates prepared from Neuro-2a cells expressing GFP fusion constructs (3 mg protein) were incubated with GFP-trap-agarose (0.05 mL) (ChromoTek). The incubation for streptavidin-agarose was overnight at 4 °C, and for GFP-trap-agarose was 2 h at room temperature. The immobilised proteins in the streptavidin- or GFP-trap-agarose pellets were thoroughly washed. For both protocols, 7 wash steps were performed, with centrifugation (13,500 × g, 1 min) between washes and with the first 2 washes using RIPA buffer. For the BioID pull-down, the 5 subsequent wash steps included: 0.5% (w/v) SDS in PBS (washes 3 and 4), 6 M urea in 100 mM TrisHCl pH 8.5 (washes 5 and 6), and 100 mM TrisHCl pH 8.5 (wash 7). For the GFP pull-down, all 5 subsequent washes used 10 mM TrisHCl pH 7.5, 150 mM NaCl, 0.5 mM EDTA (washes 3 to 7).

**Blot detection of biotinylated proteins**. Proteins binding to the streptavidin-agarose were eluted with concentrated SDS-sample buffer (180 mM TrisHCl pH 6.8, 30% (v/v) glycerol, 6% (w/v) SDS, 0.06% (w/v) bromophenol blue, 15% (v/v) β-mercaptoethanol). Samples were boiled (5 min, 95 °C). Eluted proteins were separated by SDS-PAGE (8% (w/v) polyacrylamide gels, 1.5 h) and transferred to polyvinylidene difluoride (PVDF) membranes (Amersham Life Science; 2 h, room temperature). Subsequent steps were performed by blocking with 1% (w/v) bovine serum albumin (BSA) in phosphate-buffered saline (PBS; 0.5 h, room temperature), and incubating with streptavidin-horseradish peroxidase (streptavidin-HRP) (Invitrogen, 1:2000; 1 h, room temperature) followed by thorough washing with Tris-buffered saline Tween-20 (TBST). The membrane was finally blocked with buffer (1% (w/v) BSA, 1% (v/v) Triton X-100 in PBS; 5 min), washed (TBST, 3 × 1 min), and visualised using an enhanced chemiluminescence detection system (ThermoFisher Scientific). Images were captured using ChemiDoc imager (Bio-Rad) operating in a single-channel protocol.

**Mass spectrometry (MS) analysis**. Proteins in the washed streptavidin- or GFP-trap-agarose pellets were further prepared for MS analysis. For streptavidin-agarose pellets, an on-bead digestion was performed by adding 20 μl trypsin-containing denaturing solution (50 mM urea, 5 mM tris(2-carboxyethyl) phosphine (TCEP) and 0.25 μg trypsin (Sigma) in 50 mM triethylammonium bicarbonate (TEAB)). After incubation (overnight, 37 °C, end-over-end rotation), supernatants were collected. For the GFP-trap-agarose pellets, an in-solution digestion for the eluted proteins was performed by adding the elution solution (50% aqueous 2,2,2-Trifluoroethanol (TFE)/0.05% formic acid, pH 2.0, 1 mM TCEP). After incubation (5 min, room temperature), supernatants were collected and then 40 μL trypsin solution (0.25 μg trypsin, 200 mM TEAB) added before incubation (overnight, 37 °C, end-over-end rotation).

For each trypsin-digested sample, supernatant (10 μl) was collected and tryptic peptides were analysed by liquid chromatography-MS/MS (LC-MS/MS) using an Q-Exactive plus mass spectrometer (Thermo Scientific) fitted with nanoflow reversed-phase-HPLC (Ultimate 3000 RSLC, Dionex). The nano-LC system was equipped with an Acclaim Pepmap nano-trap column (Dionex – C18, 100 Å, 75 μm × 2 cm) and an Acclaim Pepmap RSLC analytical column (Dionex – C18, 100 Å, 75 μm × 50 cm). Typically for each LC-MS/MS experiment, 1 μL supernatant was loaded onto the enrichment (trap) column (isocratic flow 5 μL/min, 3% CH₃CN containing 0.1% formic acid, 6 min) before the enrichment column was switched in-line with the analytical column. For LC, the eluents used were 0.1% formic acid (solvent A) and 100% CH₃CN/0.1% formic acid (solvent B) with the following sequence of gradients: 3–20% B (in 95 min), 20–40% B (in 10 min), 40–80% B (in 5 min), 80% B (maintained for the final 5 min) before equilibration in 3% B prior to the next analysis (10 min). All spectra were acquired in positive mode with full scan MS1 scanning from m/z 375–1400 (70000 resolution, AGC target 3e6, maximum accumulation time 50 ms, lockmass 445.120024). The 15 most intense peptide ions with charge states ≥2–5 were isolated (isolation window 1.2 m/z) and fragmented with normalised collision energy of 30 (35000 resolution, AGC target 1e5, maximum accumulation time 50 ms). An underfill threshold was set to 2% for triggering of precursor for MS2. Dynamic exclusion was activated for 30 s.

**Bioinformatic analysis of mass spectrometry data**. MS/MS data was analysed using the Mascot search engine (Matrix Science version 2.4) against the SWISSPROT database (July 2015 release; 549,008 entries) as described[14]. Briefly, the following settings were applied: taxonomy - Mus., enzyme - Trypsin, Protein Mass - ± 20 ppm, Fragment Mass Tolerance - ± 0.2 Pa, Max Missed Cleavages: 2. Test and background/non-specific binding identifications were accepted for proteins with at least two significant peptides (p < 0.05). Background/non-specific binding was defined from samples prepared from GFP only-transfected cells for the Pulldown protocol assessments, or from untransfected cells for the BioID assessments. Proteins identified in at least three of the four tested conditions (BioID ± arsenite; Pulldown ± arsenite) were retained and considered further. Ingenuity pathway analysis (IPA version 01-04; QIAGEN) was carried out according to the supplier's instructions with default settings.

**Cell death assessment**. Neuro-2a cells (2 × 10⁵ cells/well) transfected to express GFP-ataxin-1[30Q], GFP-ataxin-1[85Q], the nuclear localisation defective GFP-ataxin-1 K772T, or GFP alone as a control, were suspended in PBS and transferred to 5 ml polystyrene round-bottom tubes (BD Bioscience). To determine the numbers of dead cells per sample, the SYTOX Red dead cell stain (ThermoFisher Scientific) was added to cell suspensions, and incubated at room temperature for 15 min before flow cytometry analysis according to the manufacturer's instructions. All samples were prepared and measured in triplicate, with n = 3 independent experiments performed.

Fluorescence detection and recording was assessed by flow cytometry (BD LSRFortessa (BD Bioscience)) with the background fluorescence control (i.e. fluorescence counts from the green channel = 0–0.3 K) estimated from using untransfected cells and negative controls determined by the GFP-only cells (GFP, Green channel) and SYTOX-only cells (SYTOX, Red channel). The voltage applied to each channel was adjusted according to the GFP-only or SYTOX-only controls to avoid the fluorescence count exceeding 100 K. For analysis, 50,000–75,000 cells were recorded by the cytometer for each sample. FlowJo software (Version vX.0.7) was used in the analysis. The gating of debris and intact Neuro-2a cells was based on forward scatter-area (FSC-A) and the side scatter-area (SSC-A). The gating for transfected cells was based on fluorescence FITC-A (GFP, Green channel); the gate for dead cells was based on fluorescence APC-A (SYTOX, Red channel). For analysis to compare toxicity of GFP-ataxin-1[30Q] with GFP-ataxin-1[85Q] expressed at different levels within the cell population, further gating was applied to the transfected cells based on different fluorescence counts from the green channel: 0.3–1 K, 1–10 K, 10–100 K. Cell death percentage (%) in each fluorescence range was determined using the following formula: (number of dead cells/number of total cells) x 100.

**SCA1 mouse models**. All mice were housed and managed by Research Animal Resources (RAR) under specific pathogen–free conditions in an Association for Assessment and Accreditation of Laboratory Animal Care International

(AAALAC)–approved facility. RAR services such as feeding, housing, husbandry, and disposal is provided by a team of trained laboratory technicians led by a veterinarian. Procedures for minimising pain and discomfort were developed and undertaken in consultation with veterinarians. Ethics oversight and animal use protocol approvals were provided by the University of Minnesota Institutional Animal Care and Use Committee (IACUC).

To evaluate altered nuclear transport in the presence of polyQ-ataxin-1 in vivo, transgenic mice that express *ATXN1*[82Q] under the control of a Purkinje cell-specific promoter were analysed; these animals develop motor performance deficits and Purkinje cell degeneration that are typical features of SCA1[67]. 10–12 week old *ATXN1*[82Q] mice were analysed in parallel with FVB non-transgenic controls. Two additional ataxin-1 lines, in which ataxin-1 toxicity is ameliorated, were also analysed as controls: *ATXN1*[82Q] V591A S602D mutated to be CIC-interaction-deficient[42] (herein termed *ATXN1*[82Q]CIC(mVS)) and a shorter polyQ-length ataxin-1 but with the additional phosphomimetic mutation S776D that induces SCA1 disease initiation and Purkinje cell dysfunction but without late-stage Purkinje cell death[23]. All transgenic mice were maintained on a FVB background, with genotyping performed using standard approaches[23,42]. Briefly, transgenic mice were genotyped from genomic DNA made from ear punch tissue. PCR was performed with primers spanning a junction fragment unique to the human transgene. 1 μM of Forward 5'-AGGTTCACCGGACCAGGAAGG-3' and Reverse 5'-AATGAACTGGAAGGTGTGCGGC-3' primers were used with Promega GO Taq (cat #M3001) at an annealing temperature of 60 °C.

**Immunofluorescence, image acquisition and image analysis.** In cultured cell studies assessing protein localisation, Neuro-2a cells were cultured on coverslips (Proscitech), transfected and treated with 300 μM arsenite, as indicated. Subsequent processing (washing, permeabilization (0.2% (v/v) Triton X-100), fixation (4% (w/v) paraformaldehyde) and blocking (1% (w/v) bovine serum albumin (BSA)) was performed at room temperature in phosphate-buffered saline. Cells were then incubated sequentially with primary antibodies then fluorophore-conjugated secondary antibodies or streptavidin as indicated, each in 1% (w/v) BSA in PBS for 1 h at room temperature. The primary antibodies (1:100 dilutions) used were as follows: anti-cMyc (sc-40, Santa Cruz); anti-nucleolin (ab22758, abcam); anti-RanGAP1 (ab2081, abcam); anti-RCC1 (sc-1162, Santa Cruz)). Subsequent detection used Alexa Fluor® 568-conjugated secondary antibodies (A-11004, Invitrogen, 1:300 dilution) or Alexa Fluor® 488-conjugated Streptavidin (S11223, Invitrogen, 1:300 dilution). Processed coverslips were then mounted by water-based Fluoro-Gel (Proscitech) onto glass slides for visualisation by confocal laser scanning microscopy (Leica TCS SP5 with 63× 1.4 oil immersion objective).

All imaging studies were performed on at least three independent occasions, then the acquired images were quantitatively analysed using Fiji software (version 2017-05-30 for Mac). Quantitative analysis of nuclear body size and the fluorescence intensity of nucleus/cytoplasm ratio (Fn/c) performed using CellProfiler cell image analysis software (version 2.1.1 for Mac). In the Fn/c analyses, after applying the "CorrectIllumination" function, cytoplasmic areas were defined by subtracting the area stained with DAPI. Pearson's correlation coefficients, as a quantitative analysis of protein colocalisation/mis-localisation, were assessed by the degree of correlation of localisation between fluorophores in a 25 μm² square centred over each arsenite-induced ataxin-1 nuclear body, and were calculated using the coloc2 plugin. A higher Pearson correlation coefficient corresponds to greater colocalisation of two fluorophores, with values of 0.6 to 1.0 corresponding to higher to complete levels of co-localisation[68].

For the assessment of ataxin-1[85Q] association with selected binding partners in cells, we exploited the biomolecular fluorescence complementation (BiFC) system that is based on protein interactions bringing together ectopically expressed Venus N-terminal (VN-) and Venus C-terminal (VC-) fragments to reconstitute fluorescence of the Venus protein[38]. The BiFC system enables detection of interactions with minimal perturbance to the cells, requires no assumptions about the accessibility of the complex to extrinsic fluorophores, and is sufficiently sensitive to enable analysis of interactions between proteins expressed at levels comparable to endogenous proteins[39]. Importantly, BiFC analysis allows the detection of interactions that may involve only a subpopulation of a particular protein, which can specifically interact with many cellular proteins[41].

Thus, we cultured HeLa cells on coverslips, transfected to express the indicated BiFC constructs (VN- and VC-), and stained with 0.8 μg/ml Hoechst 33342 (Sigma Aldrich) prior to imaging by confocal laser microscopy (FluoView™ FV1000 confocal microscope with 60× water immersion objective). To ensure our system robustly reported intracellular protein-protein interactions for high levels of ectopically expressed proteins, we included the controls of the interaction of c-Fos and c-Jun and its disruption for the mutant c-Fos(Δzip)[69],and the interaction of CIC and ataxin-1[85Q] and its disruption for the mutant ataxin-1[85Q] mVS[42], as well testing the interactions between wild-type, mVS mutant, or the nuclear localisation defective K772T mutant ataxin-1[85Q] with Impα2 expressed as either wild-type, an NLS-binding deficient mutant[43] (importin-α2 D192K/E396R, abbreviated herein as importin-α2 mDE) or a mutant lacking the importin-β-binding domain within its N-terminus[44] (importin-α2 with residues 70–529 deleted, abbreviated herein as importin-α2 (ΔIBB)). Quantitative analysis of the Venus fluorescence intensity reconstituted by molecular complementation was performed (ImageJ 1.52n public domain software) each in three independent

experiments. In these analyses, nuclei were defined by Hoechst channel using the threshold command. Nuclear fluorescence, corrected for background in the absence of BiFC expression, was then determined for 80-200 cells/sample.

For the analysis of altered nuclear transport in vivo, mice (~12 weeks old, 3–4 animals per genotype) were deeply anaesthetised and transcardially perfused with 10% (v/v) buffered formalin and 50 μm sagittal sections were cut on a vibratome. Epitopes were unmasked by boiling (3 × 15 s) in 0.01 M urea. The sections (3 sections per animal) were blocked (1 h) in 2% (v/v) normal donkey serum, 0.3% (v/v) Triton X-100 in phosphate-buffered saline, then incubated (48 h, 4 °C in blocking buffer) with a range of anti-importin or anti-NUP antibodies to test their suitability for use in this tissue staining protocol. The following monoclonal antibodies were chosen for quantitative analyses as their staining patterns showed appropriate specificity and localisation patterns in control brain sections: NUP62 (BD Biosciences #610498; 1:500 dilution), NUP98 (Santa Cruz #SC-74553; 1:500 dilution) or IMPβ1 (Santa Cruz #SC-137016; 1:500 dilution) along with anti-ataxin-1 (12nq)[67] (1:2000 dilution) and anti-calbindin (Santa Cruz #SC-7691; 1:250 dilution). The sections were washed (4×) in phosphate-buffered saline. Subsequent detection used secondary antibodies (Jackson Immunoresearch; 1:500 dilution, 48 h in blocking buffer) as follows: anti-mouse Alexa Fluor® 488-conjugated (#715-545-150), anti-rabbit Cy3 (#711-165-152) and anti-goat Alexa Fluor® 647-conjugated (#705-605-147). The sections were again washed (3×) in phosphate-buffered saline then DAPI (0.1 μg/ml) was included in a final wash (15 min) before mounting with 4 mg/ml N-propyl gallate/glycerol gelatin as a fluorescence preservative. Processed samples were visualised using an Olympus Fluoview 1000 IX2 inverted microscope. Stacks of images were taken (20 μm stack at 1 μm/slice) to ensure clearest nuclear rim staining was used for each measurement and a minimum of 25 cells were analysed per section An observer blinded to the animal genotype then undertook quantitative analysis of the altered nuclear rim IMPβ1 fluorescence intensity (30–60 cells/animal, 100–180 cells total per group) was undertaken using Fiji software and the following criteria: normal-mild= strong nuclear rim staining indicated by 2~3 fluorescence peaks corresponding to nuclear rim by line scan measurement by Fiji, moderate-severe = weak nuclear rim staining indicated by irregular fluorescence peak distribution across the nucleus. Similarly, an observer blinded to the animal genotype undertook quantitative analysis of altered morphologies of nuclear rim staining for NUP98 and NUP62 (50–100 cells/animal, 180–370 cells total), using Fiji software and the following criteria: normal-mild = round to mostly round with < 2 small to medium-sized wrinkles, moderate-severe = elongated with large invaginations and distorted morphology.

**Analytical ultracentrifugation.** For the analysis of aggregate formation by ataxin-1[85Q] or ataxin-1[85Q]S776A, Ad293 were transfected to express GFP-ataxin-1 [85Q] or GFP-ataxin-1[85Q]S776A or left untransfected as a control. After 24 h, lysates were then prepared for analytical ultracentrifugation (AUC) in lysis buffer [20 mM Tris HCl pH 8.0, 2 mM MgCl₂, 1% (v/v) Triton X-100 supplemented with complete protease inhibitor mix (Roche Diagnostic) and 20 units/ml Benzonase (Novagen)]. Cell lysates were mixed by pipetting, dissociated by passing through a 27 Gauge needle at least 25 times on ice, with NaCl then added to a final concentration of 150 mM. prior to snap freezing in liquid nitrogen and storage at −80 °C until analysis. Immediately prior to analysis, lysates were thawed on ice, total cellular protein concentrations adjusted to equal levels using the lysates from untransfected cells, and the viscosity of the samples increased by the addition of sucrose to 2 M final concentration. Further steps were performed as per the protocol in[70] within the Macromolecular Interactions Facility (University of Melbourne)[70]. Briefly, samples were analysed using an XL-A analytical ultracentrifuge (Beckman Coulter, Fullerton, CA) equipped with a Fluorescence Detection System (Aviv Biomedical, Lakewood, NJ). Protein samples were loaded into double-sector epon centerpieces and placed in an AnTi 50 rotor (mim radius = 7.2 cm). Radial fluorescence data was acquired at 10 °C using a rotor speed of 724 g and an excitation wavelength of 488 nm in continuous scanning mode. Under these centrifugation and solvent conditions, only aggregates will sediment. The percentage of the aggregates was estimated from the proportion of the fast-moving sedimenting boundaries in the radial scans.

**Statistics and reproducibility.** Graphpad Prism 6 (version 6.00 for Mac) was used for statistical analysis. Data sets were analysed by D'Agostino-Pearson Omnibus normality test before further comparison analysis. For analysis of >2 datasets, ANOVA was applied to compare differences among different datasets showing normal distribution, followed by Tukey's multiple comparisons test (*$p \leq 0.05$, **$p \leq 0.01$, ***$p \leq 0.001$, ****$p \leq 0.0001$). For datasets with deviation from a normal distribution, Mann–Whitney and Kruskal–Wallis non-parametric test was used for analysis (##$p \leq 0.01$, ####$p \leq 0.0001$). Data are represented as mean ± standard error of the mean (SEM), with $p < 0.05$ regarded as statistically significant.

Each experiment was repeated three times independently with similar results. All images shown are representative results from biological replicates.

**Reporting summary.** Further information on research design is available in the Nature Research Reporting Summary linked to this article.

## Data availability

The authors declare that the datasets generated during and/or analysed during the current study are available via ProteomeXchange with identifier PXD010352[https://www.ebi.ac.uk/pride/archive/projects/PXD010352]. All other data supporting the findings of this study are available within the Article, its Supplementary Information or from the corresponding author on reasonable request. The source data underlying Figs. 2a–d, 3e, 4b,c, 5e–g, 6c and 7b–d and Supplementary Figs. 1b, 2b, 4b,c, 5c, 6d,e, 7e, 8b, 9c and 10d,e, are provided as a Source Data File.

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

## Acknowledgements

This work was funded by a grant (1121907) to M.A.B and D.A.J. from the Australian National Health and Medical Research Council (NHMRC), and an NIH/NINDS grant to H.T.O. (NS022920-30). S.Z. also acknowledges the scholarship support by the University of Melbourne (Melbourne International Research Scholarship). The mycBioID vector (pcDNA3.1 mycBioID) was a gift from K. Roux (addgene plasmid # 35700), the BiFC vectors were a gift from C.-D. Hu (addgene plasmids #22011, #22013, #22014, # 27097, #27098), and the GFP-ataxin-1[85Q] and GFP-ataxin-1[30Q] plasmids were provided by D.M. Hatters (University of Melbourne). We acknowledge access to the following facilities at the University of Melbourne: the Bio21 Institute Mass Spectrometry and Proteomics Facility, the Macromolecular Interactions Facility, Melbourne Brain Centre (MBC) flow cytometry facility, and the Biological Optical Microscopy Platform. We also acknowledge access to the Monash Micro Imaging Facility at Monash University, Clayton.

## Author contributions

S.Z., D.A.J. and M.A.B conceived the work and designed the experiments. S.Z. and N.A.W. performed the proteomics experiments, S.Z. performed the cell biology experiments, S.Z. prepared protein samples and Y.F.M. performed the AUC experiments, A.L. optimised and performed the BiFC analyses, L.D., A.K.-D and P.Y. performed the SCA1 mice analyses under the supervision of H.T.O., and all authors analysed the data. S.Z. drafted the manuscript, and all authors contributed to discussions and reviewed the manuscript.

## Competing interests

The authors declare no competing interests.
