## [Peer Review File · Nature Communications]

Reviewers' comments:

Reviewer #1 (Remarks to the Author):

The ataxin-1 interactome reveals direct connection with multiple disrupted nuclear transport pathways

In this manuscript the authors have identified direct and proximal partners of ataxin-1 in Neuro-2a (N2A) cells using overexpression of GFP-Ataxin1[85Q] or MBI-Ataxin-1[85Q], under control conditions and in response to arsenite. Pathway analysis indicated enrichment of nuclear transporters in the interactome. Authors then used imaging of nuclear transporters and their cargos to provide evidence of re-localization of transporters into ataxin-1[85Q] nuclear bodies and disrupted trafficking.

Nuclear transport is essential for cellular functioning and there have been several recent reports indicating that nucleocytoplasmic compartmentalization is impaired in several models of repeat expansion disorders (Chapple et al., 2008; Mapelli et al., 2012; Freibaum et al., 2015; Jovičić et al., 2015; Kaneb et al., 2015; Liu et al., 2015; Rodríguez et al., 2015; Zhang et al., 2015; DaCruz and Cleveland, 2016; Lee et al., 2016; Nekrasov et al., 2016; Woerner et al., 2016; Zhang et al., 2016; Shi et al., 2017, Gasset-Rosa et al., 2017, Grima et al., 2017)). Thus this is an interesting and timely topic to explore in SCA1, which is polyglutamine repeat disorder. However, there are major concerns that need to be addressed before this manuscript can be accepted for publication.

1. It is unclear what the relevance of arsenite caused stress is to the SCA1 and to what extent the changes caused by arsenite in cells expressing mutant ataxin-1 should be considered a part of polyQ pathological process? Unless clearly justified by authors, use of arsenite makes interpretation of the data more difficult.

2. All the experiments authors have presented in the manuscript are performed in vitro in N2A cells and in most cases proteins studied were overexpressed. Authors need to demonstrate similar changes in trafficking and mislocalization in the cerebellum as well as in control unaffected brain regions in the mouse model/s of SCA1. Adding the similar analysis in the cerebella of SCA1 patients would increase the impact of this study.

3. In addition to qualitative changes seen in their figures, authors should provide some quantification –for example in vivo they could determine what percentage of brain cells demonstrate changes in the localization of nuclear transport proteins. Another method such as

western blotting of nuclear and cytoplasmic fractions –would strengthen the finding that nuclear localization is altered by mutant ataxin-1.

4. Since authors are overexpressing mutant ataxin-1 [85Q] –some of the interactions may be due to non-physiologically high levels of ataxin-1 and not due to polyQ expansion. Thus wild-type murine Ataxin-1[2Q] or a human ataxin-1[30Q] should be used as controls in addition to GFP.

There are several moderate to minor concerns as well that need to be addressed:

1. Even though the images showing increase in inclusion size upon arsenite exposure are convincing, it would be helpful that the authors present a quantification of nuclear inclusion size. Do authors know how can 1hr of arsenite exposure lead to such a big change in inclusion size?

2. GFP-Ataxin1[85Q] and MBI-Ataxin-1[85Q] look very different both without and with arsenite exposure so it is unclear how can they be used interchangeably. For example in most figures throughout the manuscript (e.g. Figs., 3, 4, 57) without arsenite MBI-Ataxin-1[85Q] is not forming inclusions, while GFP-Ataxin1[85Q] is. Therefore, if inclusions are required to sequester the proteins involved in nucleocytoplasmic transport, then those pathways may not be similarly affected by arsenite in MBI-Ataxin-1[85Q] and GFP-Ataxin1[85Q]. Also of importance, most of the imaging results was performed using MBI-Ataxin-1[85Q], that as mentioned above does not seem to form inclusions without arsenite. Those experiments need to be confirmed with GFP-Ataxin1[85Q] and control GFP-Ataxin-1[2Q] or ataxin-1[30Q].

3. In Fig 4. The authors used ANOVA to determine arsenite and or Ataxin-1[85Q]'s effect on conventional importin- α/β 1-recognised cargo nuclear/ cytoplasmic fluorescence intensity, yet the data shown seems more suited for a non-parametric analysis due to its non-normal distribution. It may be interesting for the authors to determine how ATXN1[85Q] intensity correlated with the Fn/c ratio as well. Moreover, images presented are not representative of histogram results: e.g. GFP-NLS-Bgal seems to be much more cytoplasmic in arsenite+ MBI-ATXN1[85Q] than in MBI-ATXN1[85Q] only, but histogram seems to show the opposite. Can authors clarify this discrepancy? Moreover, authors should similarly quantify cargo mislocalization in Figs 5C and D.

4. Images need to be similarly exposed when presenting co-localization or the lack of it. For example, in Fig. 3D GFP control seems much more exposed than any of the GFP tagged proteins in Fig. 3A-C, and overexposure can mask any co-localization.

5. This is also true for Fig 7. To show that ATXN1 causes mislocalization of NUP98, depending on the GLFG domain of NUP98, the authors show images of different truncations of NUP98, yet they claim the only enrichment is seen with GLFG (221-504) which is also the least exposed/expressed. Therefore, to more clearly show co-localization-it would be beneficial if the authors presented the orthogonal views of the images as well.

Reviewer #2 (Remarks to the Author):

The manuscript “The ataxin-1 interactome reveals direct connection with multiple disrupted nuclear transport pathways” presents an interactome analysis of ataxin-1[85Q]. The authors find that several components of the nuclear transport machinery re-localize to ataxin-1 nuclear bodies. They suggest that disruption of the nuclear transport system could ultimately disturb cellular homeostasis. There is a number of concerns that should be addressed prior to publication.

Major points:

It is not surprising that nucleoporins (and the transport receptors they bind) co-localize that with aggregates, although it is exciting to see this also for ataxin-1 nuclear bodies. The authors still see a good rim staining after induction of stress. Is it thus fair to use the term ‘re-localize’ or could it be that newly synthesized nucleoporins go to these bodies and the ‘old’ NPCs remain intact? This way in indirect effect could be induced in the longer term. This particularly relevant for the experiments done with Nup98 over-expression, for which the authors never really see convincing NPC localization from the 1st place. Since they say Nup62 is not ataxin-1 but huntingtin-relevant, overall their claims regarding Nups are relatively weak (see also point 4 below).

The authors base their analysis on two different construct: GFP-ataxin-1-[85Q] and MBI(myc-BirA)-ataxin-1-[85Q]. In Figure 1 A and C the pattern of these two constructs are shown but the staining pattern is clearly different. Detection of GFP shows distinct spots but anti-Myc has a diffuse localization all over the nucleus. The authors do not comment on these different patterns even though all further experiments are based on them. It should have been investigated how these differences can be explained. Does the fusion of GFP or Myc-BirA* influence the localization? Is there a difference when the fusion is done on the C-Terminus of ataxin-1[85Q]? The immunofluorescence staining done in all further experiments are using either one of the construct. How were those choices made? Since already the initial experiment shows differences, all staining should be performed using both constructs.

The interactome of polyQ-ataxine-1 is identified by BioID and pull-down on GFP. For better comparison the authors should have performed a classical pull-down approach using the Myc-tag already present in the MBI-ataxin-1-[85Q] construct. The supplementary table 1 contains all

identified proteins. Proteins excluded should be mentioned as well. For better comparison the measured intensities (e.g. iBAQ scores) rather than pure observation (yes/no) should be mentioned. This analysis is purely qualitative but not quantitative. These datasets should be statistically analyzed separately since the identified proteins are either enriched for stable interaction (GFP PD) or unstable/transient interactions (BioID). The authors also miss to analyze the effect of arsenite stress on either interactome, which could give additional insight into the mode of action.

The following interactome analysis is based on 91 proteins identified in at least 3 out of 4 conditions. RAN signaling was identified as the top-ranked pathway. Was the interactome analysis based on a ranked list of proteins? Nuclear transporter receptors are highly abundant within the cell which might inflate the likelihood of identification.

Minor points:

1. Do other pro-oxidant stressors have the same effect or is it arsenite specific?
2. How is the change of Hikeshi (Figure 5) explained? Due to generally increased stress levels or a specific effect?
3. IMP16 is mentioned only once (Saksena et. al, 2006). It seems to be relevant for the ER to NE localization of proteins therefore a purely nuclear staining (Figure S1C) is unexpected.
4. The authors claim that the GLFG repeats of Nup98 are relevant for the colocalization to nuclear bodies upon stress. Nup98 has 4 GLFG repeats but additionally many others FG repeats (FG, SAFG, FxFG, PxFG, see Review Lemke 2016). The Nup98 GLFG truncation (221-504) contains only 3 of the 4 GLFG repeats. Mutations of the respective GLFG repeats could explain which exact GLFG repeat is responsible for the colocalization.
5. In Figure S5 the Nup62 localization is shown. Nup62 is also recognized by mAb414. Therefore a punctual localization within the nucleus should be visible in Figure 4.
6. In Figure S1 the GFP signal of IMP α 4, IMP7 and IMP16 seems to be quite strong/close to saturation and a potential change in localization not detectable.
7. Using a log scale for the integrated fluorescence intensity analysis (Figure 4, S4 and S6) would demonstrate the changes better, especially for Figure 4. In Figure S6B and D the y-axis has a different scale and gives a false impression on the actual change.
8. Are the two NLS M9 and BIB tested functional? The N-Terminus of the BIB domain is partly buried within Importin β upon binding. The authors should place GFP additionally at the C-Terminus to verify the result.
9. In Figure S6A and S6B the authors test the localization sequence M9, specific for Transportins. Additionally it should be tested whether Transportins itself is also affected by stress.

10. In the Method section the authors should mention the origin of all plasmids used, details of the database used for MS data analysis (number of entries, version of the database) and the concentration of the antibodies used.

11. Regarding the sentences 'either directly or in conjunction with IMP α 2 that acts as the adaptor': There are multiple importin alphas that act as adaptors, not only α 2.

Reviewer #3 (Remarks to the Author):

This manuscript used coIP and proximate labelling techniques, followed by mass spec analyses, which identified many nucleocytoplasmic transport components as interactors of Ataxin-1-polyQ expansion. The authors further showed that some of these proteins are mislocalized/co-localized with Ataxin-1 granules in the nuclei. It's a clean paper with convincing results that seem to come from people who know the field/literature very well. These results lead to an important conclusion to the neurodegeneration field, given the emerging literature about nuclear transport. Below are some suggestions for the authors to consider to boost the impact of their discovery.

1. In vivo studies of protein pathology. There are SCA1 mouse and fly models available and the authors are aware of them. One experiment to do is staining of the transport components in these models. It would be interesting to know whether these proteins, e.g. Nup98, is mislocalized.

2. Functional relevance. The authors may want to consider genetic interaction studies of SCA1 and transport factors. The authors are aware of the fly screen from Juan Botas's lab on SCA1, (e.g. Nup44A). It would be good to validate some of their hits as what Botas did.

3. Nucleocytoplasmic transport defects are recently identified in several neurodegenerative diseases. A 2016 Science paper by Woerner et al. suggested that cytoplasmic protein aggregates disrupt transport. Here, this paper shows that nuclear stress can also disrupt transport, which is actually not surprising. One explanation to how protein misfolding disrupts transport is that these aggregates, usually containing low complexity domains (e.g., a polyQ), affect liquid-liquid phase transition that plays a critical role in nuclear transport. Indeed, Nup98 contains FG-repeats that undergo phase transition. Furthermore, the staining of Ataxin-1 polyQ granules in the nuclei look like the nucleoli, membrane-less organelles that is assembled due to phase transition. In addition, nucleoli are RNA granules that respond to arsenite stress. So one wonders if Ataxin-1-polyQ disrupts

transport by sequestering transport factors (e.g. Nup98) in nucleoli due to impaired liquid-liquid phase transition.

But other issues that may be considered substantial include:

1. Controls. The authors use an overexpression of the ataxin 1 poly Q expansion for all of their studies. What is lacking is a comparison to OE of WT ataxin 1 without the poly Q expansion. This comparison would allow the reviewers to define what may be the result of the disease associated expansion vs what is merely due to OE of ataxin 1.

2. Reliance on OE. While the authors present solid data, they rely solely on OE models (OE of Ataxin 1 PolyQ, co-OE with importin α or Nup98 (i.e. Figure 3/7) etc.). The story would be far more convincing and made more disease relevant if the authors were to confirm key findings in a system expressing the ataxin 1 poly Q expansion at endogenous levels (i.e. -staining in human tissue).

3. Functional significance (in addition to that detailed above). What is the significance of the interaction between ataxin 1 poly Q and these nuclear/cytoplasmic transport factors? A functional assay assessing nuclear/cytoplasmic transport (i.e. FRAP) in the context of ataxin 1 poly Q should be carried out.

RESPONSE TO REVIEWERS:

NCOMMS-17-19917: The ataxin-1 interactome reveals direct connection with multiple disrupted nuclear transport pathways

Reviewer #1

R1 Comment/Question 1: Justify the use of arsenite

(1. It is unclear what the relevance of arsenite caused stress is to the SCA1 and to what extent the changes caused by arsenite in cells expressing mutant ataxin-1 should be considered a part of polyQ pathological process? Unless clearly justified by authors, use of arsenite makes interpretation of the data more difficult)

Response to R1 C/Q1:

We now include additional data highlighting the impact of arsenite to drive increased nuclear body size and ataxin-1 aggregation in cells (new Supp Fig S1). This is a pro-oxidant stress (see references #26 (Prakash et al. 2015) and #27 (Yu et al. 2014), but we now also highlight oxidative stress resulting from direct exposure to hydrogen peroxide also has impacts that parallel those observed in the presence of arsenite (new Supp Fig S13). We stress that our proteomics assessments and analysis of impact on nuclear transport events in cultured cells have been made both without and with arsenite exposure, with the clear outcome that nuclear transport disruption is a key feature of ataxin-1 expression; this observation is now extended and confirmed by our in vivo analyses (see response to R1 Comment/Question 2).

R1 Comment/Question 2: Demonstrate changes in vivo

(2. All the experiments authors have presented in the manuscript are performed in vitro in N2A cells and in most cases proteins studied were overexpressed. Authors need to demonstrate similar changes in trafficking and mislocalization in the cerebellum as well as in control unaffected brain regions in the mouse model/s of SCA1. Adding the similar analysis in the cerebella of SCA1 patients would increase the impact of this study)

Response to R1 C/Q2:

We now include additional data in the SCA1 mouse model (ataxin-1[82Q]-expressing) alongside controls of wild-type FVB animals and two additional lines in which ataxin-1 toxicity is ameliorated: ataxin-1[82Q] V591A S602D mutated to be CIC-interaction-deficient (reference #50; Rousseaux et al 2018) (termed ataxin-1[82Q]CIC(mVS)) and a shorter polyQ-length ataxin-1 but with the additional phosphomimetic mutation S776D that induces SCA1 disease initiation and Purkinje cell dysfunction but without late-stage Purkinje cell death (reference #31; Duvick et al, 2010). We describe the protocols/procedures on pages 16-17 of the revised manuscript, include the new data as revised Figures 6 and 9, and integrate these findings into our results descriptions on pages 7 & 10 of the revised manuscript. These new data reinforce our conclusion that nuclear transport disruption is a feature of pathological ataxin-1 expression.

R1 Comment/Question 3: Include quantification

(3. In addition to qualitative changes seen in their figures, authors should provide some quantification –for example in vivo they could determine what percentage of brain cells demonstrate changes in the localization of nuclear transport proteins. Another method such as western blotting of nuclear and cytoplasmic fractions –would strengthen the finding that nuclear localization is altered by mutant ataxin-1)

Response to R1 C/Q3:

We include quantitative assessments of our cell data in our revised manuscript Figures 4B, 5E & 5F, S1, S3, S9, S11, S12 and our animal data in Figures 6B & 6C and Figures 9B & 9D. In all cases, quantitative data has been evaluated for statistically significant differences; as suggested by Reviewer 2 (see R2 MC/Q7 below), all localisation data is now presented as $\text{Log}_{10}\text{Fn/c}$. These analyses are superior to those relying on blot analysis of fractionated cells/tissues that do not give any insight into the cell-to-cell heterogeneity of response.

R1 Comment/Question 4: Include ataxin-1 control with lower Q number

(4. Since authors are overexpressing mutant ataxin-1 [85Q] –some of the interactions may be due to non-physiologically high levels of ataxin-1 and not due to polyQ expansion. Thus wild-type murine Ataxin-1[2Q] or a human ataxin-1[30Q] should be used as controls in addition to GFP)

Response to R1 C/Q4:

We now include detailed analyses of the ataxin-1[30Q] protein as Supp Fig S3 (toxicity), Supp Fig S4 (proteomics), Fig S5 (localisation). The impact of ataxin-1[30Q] is comparable to that for the ataxin-1[85Q] protein, i.e. that ataxin-1[30Q] is toxic, shows an enrichment of nuclear transporters in its interactome and alters localization of importins, an NLS-cargo, and NUP98. Indeed, this is a predictable result given the previous conclusion that higher levels of wild-type ataxin-1 can be toxic in vivo (Ref #42 Gennarino et al 2015; cited in the revised manuscript page 4). Thus, we make the conclusion as per revised manuscript page 5/6, that these results are "consistent with the notion that it is the elevation of ataxin-1 levels that permits the formation of nuclear bodies and influences the distributions of these nuclear transport proteins".

R1 Minor Comment/Question 1: Nuclear body size increases upon arsenite exposure

(1. Even though the images showing increase in inclusion size upon arsenite exposure are convincing, it would be helpful that the authors present a quantification of nuclear inclusion size. Do authors know how can 1hr of arsenite exposure lead to such a big change in inclusion size?)

Response to R1 MC/Q1:

We present more background data on the impact of arsenite in this system now in revised Supp Fig S1 and described on page 3 of the revised manuscript. Specifically, we quantitate the nuclear body size (Supp Fig S1B), show fusion events that lead to increased nuclear body size (Supp Fig S1C), but also reveal by implementing analytical ultracentrifugation (AUC) analysis for the first time that there is increased aggregate formation in of ataxin-1[85Q] in the presence of arsenite (Supp Fig S1D). These events of fusion and aggregate formation are likely contributors to ataxin-1 nuclear body size. Furthermore, we also include a previously characterised ataxin-1[85Q]S776A mutant that is considered to have lowered toxicity and demonstrate a dampened impact of arsenite.

R1 Minor Comment/Question 2: MBI-ataxin-1[85Q] vs GFP-ataxin-1[85Q], GFP-ataxin-1[30Q]

(2. GFP-Ataxin1[85Q] and MBI-Ataxin-1[85Q] look very different both without and with arsenite exposure so it is unclear how can they be used interchangeably. For example in most figures throughout the manuscript (e.g. Figs., 3, 4, 5 7) without arsenite MBI-Ataxin-1[85Q] is not forming inclusions, while GFP-Ataxin1[85Q] is. Therefore, if inclusions are required to sequester the proteins involved in nucleocytoplasmic transport, then those pathways may not be similarly affected by arsenite in MBI-Ataxin-1[85Q] and GFP-Ataxin1[85Q]. Also of importance, most of the imaging results was performed using MBI-Ataxin-1[85Q], that as mentioned above does not seem to form inclusions without arsenite. Those experiments need to be confirmed with GFP-Ataxin1[85Q] and control GFP-Ataxin-1[2Q] or ataxin-1[30Q].)

Response to R1 MC/Q2:

We note that both GFP-Ataxin-1[85Q] and MBI-Ataxin-1[85Q] show nuclear localisation, consistent with the report that ataxin-1 nuclear localization is an important in SCA1 pathology in vivo (Klement et al., 1998 – reference #33, now cited on page 3 of the revised manuscript), and that both form nuclear bodies under arsenite stress conditions. That our proteomics approaches with these epitope-tagged forms of ataxin-1 are in agreement (i.e. both identify nuclear transport proteins as a part of the interactome) re-emphasises that it is not the nuclear body size/morphology that underlies an impact on nuclear transport, but that the interaction is a fundamental feature of the ataxin-1 protein and not the specific epitope tag employed to enable such analyses.

R1 Minor Comment/Question 3: Data quantification, quantitation, statistical analysis &

presentation of results

(3. In Fig 4. The authors used ANOVA to determine arsenite and or Ataxin-1[85Q]'s effect on conventional importin- β -recognised cargo nuclear/cytoplasmic fluorescence intensity, yet the data shown seems more suited for a non-parametric analysis due to its non-normal distribution. It may be interesting for the authors to determine how ATXN1[85Q] intensity correlated with the Fn/c ratio as well. Moreover, images presented are not representative of histogram results: e.g. GFP-NLS-Bgal seems to be much more cytoplasmic in arsenite+ MBI-ATXN1[85Q] than in MBI-ATXN1[85Q] only, but histogram seems to show the opposite. Can authors clarify this discrepancy? Moreover, authors should similarly quantify cargo mislocalization in Figs 5C and D)

Response to R1 MC/Q3:

We ensured that all data sets were first analysed by the D'Agostino-Pearson Omnibus normality test before further comparison analysis, and we now state this on page 18 of the revised manuscript under the section "Statistical analysis". Upon reanalysis, only data presented in Figure 4 did not show normal distributions, and so this was reanalyzed using the Mann-Whitney and Kruskal-Wallis non-parametric test with p values denoted in Figure 4 accordingly.

We now replace images in Figure 4 and 5, that were previously randomly selected, with images that more closely align with the average results presented in our quantitative analyses. In addition, the quantitation of data for Fig 5C and 5D are now presented in Fig 5E and 5F, respectively, in the revised manuscript.

R1 Minor Comment/Questions 4&5: Image exposures & Mislocalization of NUP98

(4. Images need to be similarly exposed when presenting co-localization or the lack of it. For example, in Fig. 3D GFP control seems much more exposed than any of the GFP tagged proteins in Fig. 3A-C, and overexposure can mask any co-localization)

(5. This is also true for Fig 7. To show that ATXN1 causes mislocalization of NUP98, depending on the GLFG domain of NUP98, the authors show images of different truncations of NUP98, yet they claim the only enrichment is seen with GLFG (221-504) which is also the least exposed/expressed. Therefore, to more clearly show co-localization-it would be beneficial if the authors presented the orthogonal views of the images as well)

Response to R1 MC/Q4&5:

All results we present from our cell-based experiments are acquired on a Leica SP5 microscope (that clearly highlights saturated pixels) so that we are confident that all images we present are acquired at levels below saturation. We have further refined our wording (on page 9 of the revised manuscript and in the title for Figure 8) of the interpretation of the images acquired for NUP98 (endogenous or ectopically expressed) to emphasize the distinct nuclear puncta that we observe, and more clearly describing this as a mis-localisation.

Reviewer #2

R2 Comment/Question 1 & R2 Minor Comment/Question 4: NUP98 results

(1. It is not surprising that nucleoporins (and the transport receptors they bind) co-localize that with aggregates, although it is exciting to see this also for ataxin-1 nuclear bodies. The authors still see a good rim staining after induction of stress. Is it thus fair to use the term 're-localize' or could it be that newly synthesized nucleoporins go to these bodies and the 'old' NPCs remain intact? This way in indirect effect could be induced in the longer term. This particularly relevant for the experiments done with Nup98 over-expression, for which the authors never really see convincing NPC localization from the 1st place. Since they say Nup62 is not ataxin-1 but huntingtin-relevant, overall their claims regarding Nups are relatively weak (see also point 4 below))

(4. The authors claim that the GLFG repeats of Nup98 are relevant for the colocalization to nuclear bodies upon stress. Nup98 has 4 GLFG repeats but additionally many others FG repeats (FG, SAFG, FxFG, PxFG, see Review Lemke 2016). The Nup98 GLFG truncation (221-504) contains only 3 of the 4 GLFG repeats. Mutations of the respective GLFG repeats could explain which exact GLFG repeat is responsible for the colocalization)

Response to R1 C/Q1 & MC/Q4:

To reinforce our analyses of the impact of ataxin-1 on Nup98 localisation, we have analysed samples from SCA1 mouse models – see Response to R1 Comment/Question 2.

R2 Comment/Question 2:

2. The authors base their analysis on two different construct: GFP-ataxin-1-[85Q] and MBI(myc-BirA)-ataxin-1-[85Q]. In Figure 1 A and C the pattern of these two constructs are shown but the staining pattern is clearly different. Detection of GFP shows distinct

spots but anti-Myc has a diffuse localization all over the nucleus. The authors do not comment on these different patterns even though all further experiments are based on them. It should have been investigated how these differences can be explained. Does the fusion of GFP or Myc-BirA* influence the localization? Is there a difference when the fusion is done on the C-Terminus of ataxin-1[85Q]? The immunofluorescence staining done in all further experiments are using either one of the construct. How were those choices made? Since already the initial experiment shows differences, all staining should be performed using both constructs.

Response to R2 C/Q2:

See above: Response to R1 Minor Comment/Question 2; we now include the explanation on page 12 of the revised manuscript that we consistently use GFP-tagged nuclear transport proteins in assessing the impact of polyQ ataxin-1 in the cultured cell models. Our inclusion of the data obtained from the in vivo models of SCA1, that do not rely on detection of epitope tags, reinforce the validity of these findings without the requirement to re-perform all cell experiments with different epitope tagged constructs.

R2 Comment/Question 3: Proteomics protocol

3. The interactome of polyQ-ataxine-1 is identified by BioID and pull-down on GFP. For better comparison the authors should have performed a classical pull-down approach using the Myc-tag already present in the MBI-ataxin-1-[85Q] construct. The supplementary table 1 contains all identified proteins. Proteins excluded should be mentioned as well. For better comparison the measured intensities (e.g. iBAQ scores) rather than pure observation (yes/no) should be mentioned. This analysis is purely qualitative but not quantitative. These datasets should be statistically analyzed separately since the identified proteins are either enriched for stable interaction (GFP PD) or unstable/transient interactions (BioID). The authors also miss to analyze the effect of arsenite stress on either interactome, which could give additional insight into the mode of action.

The following interactome analysis is based on 91 proteins identified in at least 3 out of 4 conditions. RAN signaling was identified as the top-ranked pathway. Was the interactome analysis based on a ranked list of proteins? Nuclear transporter receptors are highly abundant within the cell which might inflate the likelihood of identification.

Response to R2 C/Q3:

That we use completely independent epitope-tagged constructs is a strength of our approach and avoids any potential artefacts introduced in the analysis. Following the submission of our original manuscript, we were encouraged to submit an additional *Data Descriptor* manuscript to the *Nature Communications* sister journal *Scientific Data* to thus describe our dataset in a more comprehensive fashion to support data reuse. Our detailed datasets are thus now available via ProteomeXchange with the identifier PXD010352 and further details published after peer review (reference #22 in the revised manuscript: Zhang et al 2018 Sci Data 5, 180262).

R2 Minor Comment/Question 1: Impact of stressors

(1. Do other pro-oxidant stressors have the same effect or is it arsenite specific?)

Response to R2 MC/Q1:

We present new data in Supp Figure 13 of the revised manuscript showing that hydrogen peroxide also has impacts comparable to arsenite on IMP α 2, IMP β 1, the NLS-tagged cargo, and NUP98, and so the impact is not arsenite-specific. This impact of the direct application of hydrogen peroxide is described on page 10 of the revised manuscript.

R2 Minor Comment/Question 2:

(2. How is the change of Hikeshi (Figure 5) explained? Due to generally increased stress levels or a specific effect?)

Response to R2 MC/Q2:

We include Hikeshi (and its cargo HSP70) as a representative of the broader class of nuclear transporters rather than trying to link its actions specifically to stress levels. The presence of arsenite alone is not sufficient to disrupt HSP70 localisation; we show disrupted HSP70 localisation in the presence of ataxin-1[85Q] expression regardless of the presence of arsenite (Fig 5F). Thus the impacts of elevated ataxin-1[85Q] levels impact multiple, but not all, transporters.

R2 Minor Comment/Question 3:

(3. IMP16 is mentioned only once (Saksena et al, 2006). It seems to be relevant for the ER to NE localization of proteins therefore a purely nuclear staining (Figure S1C) is unexpected)

Response to R2 MC/Q3:

There is confusion here regarding importin-16 that appears to arise from the complicated IMP nomenclature used in the published literature on trafficking in different species/systems. In our studies, we evaluated mammalian importin-16 (revised manuscript Figure S6C), also known as RanBP16 or exportin-7 (names which we now include on page 12 of the revised manuscript) which is not involved in ER to NE localization of proteins. However, Saksena et al (Nat Struct Mol Biol 2006 13: 500-508) have studied IMP- α -16 from *Spodoptera frugiperda* (the insect known as the fall armyworm moth) showing it was involved in sorting membrane proteins from the ER to their ultimate nuclear envelope localization. Thus, our results for RanBP16, a protein involved in nuclear/cytoplasmic transport, are as expected.

R2 Minor Comment/Question 4:

See response above: Response to R1 Comment/Question 1 and Minor Comment/Question 4

R2 Minor Comment/Question 5:

(5. In Figure S5 the Nup62 localization is shown. Nup62 is also recognized by mAb414. Therefore a punctual localization within the nucleus should be visible in Figure 4)

Response to R2 MC/Q5:

Images acquired following staining with anti-NUP62 antibodies and anti-NPC antibodies (mAb414) are presented in the revised manuscript as Supp Figure 10 and Figure 6, respectively. The patterns of staining we observe under control conditions are consistent with published literature; we now include references in the revised manuscript to support this (references #53 & #58 in the revised manuscript).

R2 Minor Comment/Question 6:

(6. In Figure S1 the GFP signal of IMP?4, IMP7 and IMP16 seems to be quite strong/close to saturation and a potential change in localization not detectable)

Response to R2 MC/Q6:

See above: R1 MC/Q4&5.

R2 Minor Comment/Question 7:

(7. Using a log scale for the integrated fluorescence intensity analysis (Figure 4, S4 and S6) would demonstrate the changes better, especially for Figure 4. In Figure S6B and D the y-axis has a different scale and gives a false impression on the actual change)

Response to R2 MC/Q7:

We now present all quantitative localisation data as $\log_{10}Fn/c$: in the revised manuscript (Figures 4B, 5E&F, Supp Fig S9B, Supp Fig S11B&D, Supp Fig S12B). In the revised version of S6B & D (new Figure S11 B & D), we present the data on the same scale.

R2 Minor Comment/Question 8: NLS constructs functional

(8. Are the two NLS M9 and BIB tested functional? The N-Terminus of the BIB domain is partly buried within Importin ? upon binding. The authors should place GFP additionally at the C-Terminus to verify the result)

Response to R2 MC/Q8:

The use of GFP-M9-NLS and GFP-BIB-NLS has been previously reported (reference #64: Gustin et al 2001; reference #63: Jakel et al 1998). For both, F_n/c values are >1 (F_n/c GFP-M9-NLS = $1.97 + 0.13$; F_n/c GFP-BIB-NLS = $41.9 + 4.8$) thus confirming their functionality without need to alter the position of the GFP tag; this is now included on page 9 of the revised manuscript. As per the response to Question 9 below, we demonstrate altered transportin localisation that additionally supports the conclusion that M9 transport can be disrupted by ataxin-1 expression.

R2 Minor Comment/Question 9:

(9. In Figure S6A and S6B the authors test the localization sequence M9, specific for Transportins. Additionally it should be tested whether Transportins itself is also affected by stress)

Response to R2 MC/Q9:

We now include data on transportin-1 (TNPO1) (revised manuscript Supp Figure S12, and pages 9-10). Quantitative image analysis (Supp Figure S12) confirms its mis-localisation in the presence of ataxin-1.

R2 Minor Comment/Question 10:

(10. In the Method section the authors should mention the origin of all plasmids used, details of the database used for MS data analysis (number of entries, version of the database) and the concentration of the antibodies used)

Response to R2 MC/Q10:

We now include these additional experimental details as follows:

Origin of all plasmids – revised manuscript page 12

Details of the database – revised manuscript page 15

Concentrations of the antibodies – revised manuscript pages 16-17

R2 Minor Comment/Question 11:

(11. Regarding the sentences 'either directly or in conjunction with IMPalpha2 that acts as the adaptor': There are multiple importin alphas that act as adaptors, not only alpha2)

Response to R2 MC/Q11:

We have improved the accuracy of this sentence by ensuring we refer to “an” (not “the”) adaptor.

Reviewer #3

R3 Comment/Questions 1 & 2):

(1. In vivo studies of protein pathology. There are SCA1 mouse and fly models available and the authors are aware of them. One experiment to do is staining of the transport components in these models. It would be interesting to know whether these proteins, e.g. Nup98, is mislocalized)

(2. Functional relevance. The authors may want to consider genetic interaction studies of SCA1 and transport factors. The authors are aware of the fly screen from Juan Botas's lab on SCA1, (e.g. Nup44A). It would be good to validate some of their hits as what Botas did)

Response to R3 C/Qs 1 & 2:

See above: Response to R1 Comment/Question 2, in which we have extended our in vitro observations to analyses in the SCA1 mouse model together with 2 additional lines in which SCA1 toxicity is ameliorated. We acknowledge the interesting work in *Drosophila* screening in our manuscript (page 10) by referring to the work on the *Drosophila* SCA1 model that previously identified the nucleoporin NUP44A as a suppressor of toxicity (reference #66: Fernandez-Funez et al 2000; reference #67: Branco et al 2008 as well as the additional analyses for suppressors of C9orf72 repeat expansion toxicity in the *Drosophila* ALS model, that identified multiple nuclear pore complex proteins, importins/exportins and Ran regulators (reference #25: Boeynaems et 2016; reference #13: Freibaum et al 2015; reference #14: Zhang et al 2015).

R3 Comment/Question 3:

3. Nucleocytoplasmic transport defects are recently identified in several neurodegenerative diseases. A 2016 Science paper by Woerner et al. suggested that cytoplasmic protein aggregates disrupt transport. Here, this paper shows that nuclear stress can also disrupt transport, which is actually not surprising. One explanation to how protein misfolding disrupts transport is that these aggregates, usually containing low complexity domains (e.g., a polyQ), affect liquid-liquid phase transition that plays a critical role in nuclear transport. Indeed, Nup98 contains FG-repeats that undergo phase transition. Furthermore, the staining of Ataxin-1 polyQ granules in the nuclei look like the nucleoli, membrane-less organelles that is assembled due to phase transition. In addition, nucleoli are RNA granules that respond to arsenite stress. So one wonders if Ataxin-1-polyQ disrupts transport by sequestering transport

factors (e.g. Nup98) in nucleoli due to impaired liquid-liquid phase transition)

Response to R3 C/Q3:

We now include analysis of anti-nucleolin staining for control cells and those expressing either GFP-ataxin-1[85Q] or MBI-ataxin-1[85Q] under control and arsenite stress conditions (Fig S2 in the revised manuscript), defining negligible overlap of staining and thus concluding as per page 3 of the revised manuscript that these are different nuclear structures.

R3 Minor Comment/Question 1: ataxin-1[30Q] control

(1. Controls. The authors use an overexpression of the ataxin 1 poly Q expansion for all of their studies. What is lacking is a comparison to OE of WT ataxin 1 without the polyQ expansion. This comparison would allow the reviewers to define what may be the result of the disease associated expansion vs what is merely due to OE of ataxin 1)

Response to R3 MC/Q1:

As addressed in Response to R1 Comment/Question 4: higher levels of wild-type ataxin-1 indeed cause pathology, and consistent with this important observation we note that ataxin-1[30Q] also includes multiple nuclear transporters in its interactome.

R3 Minor Comment/Question 2: Use of an in vivo disease model

(2. Reliance on OE. While the authors present solid data, they rely solely on OE models (OE of Ataxin 1 PolyQ, co-OE with impotin a or Nup98 (i.e. Figure 3/7) etc.). The story would be far more convincing and made more disease relevant if the authors were to confirm key findings in a system expressing the ataxin 1 poly Q expansion at endogenous levels (i.e. -staining in human tissue)

Response to R3 MC/Q2:

See above: Response to R1 Comment/Question 2, in which we explain our new analyses on SCA1 mice, including assessment of two additional control lines with ameliorated pathology.

R3 Minor Comment/Question 3: Assessed impact on nuclear transport

(3. Functional significance (in addition to that detailed above). What is the significance of the interaction between ataxin 1 polyQ and these nuclear/cytoplasmic transport factors? A functional assay assessing nuclear/cytoplasmic transport (i.e. FRAP) in the context of ataxin 1 polyQ should be carried out)

Response to R3 MC/Q3:

The functional impact on nuclear transport has been evaluated by quantitative analysis of the nuclear/cytoplasmic distributions of multiple cargo proteins (Fig 4: NLS- β -gal; Fig 5: NFYB & HSP70; Supp Fig S9: AF-10; Supp Fig S11: M9-NLS & BIB-NLS).

Reviewers' comments:

Reviewer #1 (Remarks to the Author):

Authors have addressed several of the concerns improving the revised manuscript. Most importantly they included analysis of transgenic SCA1 model and human wt ATXN1[30Q] as a control, as well as more quantification of their results.

While these changes improved the manuscript, authors have also significantly decreased the importance/novelty of the manuscript by already publishing some of the results including the ataxin-1 interactome in their recent paper (Zhang et al, Sci Data 5, 180262, 2018).

In addition there are several major concerns that need to be addressed.

1. Authors need to validate their interactome data, for at least some of their top five interactors (e.g. from Supplementary table 3). For example they could test ATXN1 binding of the identified ataxin-1 interacting partners through Co-IPs from the brain extracts.

2. Authors have found that overexpressing non-pathogenic WT ATXN1[30Q] in non-differentiated N2A cells causes similar disruption of nuclear transport as does overexpression of pathogenic expanded ATXN1[85Q]. Authors conclude that wild-type ATXN1[30Q] is toxic and that "Ataxin-1 protein levels and cellular toxicity are correlated", thus just increasing ATXN1 expression should cause these disruptions in nuclear transport. To support this they state that overexpression of wt ATXN1 can cause neurodegeneration (citing Gennarrino et al 2015 that showed that Pumilio haploinsufficiency upregulates ATXN1 expression (using Pumilio +/- mice), and causes cerebellar neurodegeneration). It is important for authors to note that in addition to increased ATXN1 expression, Pumilio +/- mice may have other, non-ATXN1 related features. Importantly, they should note that AO2 mice that overexpress ATXN1[30Q], are indistinguishable on rotarod from wild-type controls at 1 year of age (Clark et al. J Neuroscience 1997). Finally, to directly correlate nuclear transport disruptions to ATXN1 pathogenicity in vivo, authors should examine whether nuclear transport is disrupted in AO2 mice.

3. For their in vivo studies authors used transgenic mouse line ATXN1[82Q] that overexpresses mutant ATXN1 30-60 times. They also analyzed ATXN1[30Q] D776 mice that express phosphomimetic ATXN1[30Q] and demonstrate Purkinje neuron dysfunction, cerebellar pathology and ataxia, and ATXN1[82Q] V591A S 602D mice expressing ATXN1[82Q] that cannot bind capicua and has significantly ameliorated phenotype. Intriguingly, authors claim that ATXN1[82Q] line demonstrates nuclear transport disruption (based on IMPB1 and NUP62), while ATXN1[30Q] D776 and ATXN1[82Q] V591A S602D do not. Since all three lines show accumulation of ATXN1 in the nucleus could this

suggest that it is not overexpression of ATXN1 that causes these changes (as claimed above)? And if it is direct interaction of ATXN1 with nuclear transport proteins that causes disruption, authors should comment why is this abrogated when ATXN1 cannot bind Capicua?

4. Most of the results (e.g. Figs 3, 4, 5, 7, 8, SF2) are derived from two independent experiments. It is our opinion that to ensure reproducibility minimum of three independent experiments is required for any publication. Additionally, some of these figures are still lacking quantification (Fig 3, 7, 8, SF2).

5. Could authors please explain the use of the two different descriptive measures of nuclear transport disruptions in mice in Figs 6 (number or irregularity of the peaks) and 9 (shape of the nucleus)? It would be much more informative if authors would use the same quantifiable and objective measure for both figures. For example for both figs 6 and 9 authors could use % change in fluorescence from the rim to the inside of the nucleus?

6. With the exceptions of Figs 1 and 2, authors use untransfected cells as controls in most of their experiments. As transfection reagents, DNA, as well as overexpression of proteins, can all stress and alter cells, appropriate controls for those experiments with transfection and overexpression of MBI ATXN1 or GFP ATXN1 would be transfections with MBI or GFP vectors.

Minor issues:

1. The focus and depth of imaging can both influence nuclear rim staining. Could authors please describe imaging parameters they used to control for this?

2. Can authors please comment on colocalization of GFP-NLS-B gal and MBI-ATXN1 in Fig 4A?

3. Could authors please describe how did they define cytoplasm for their Fn/c quantification?

4. Is lack of cytoplasmic IMPB1 staining in Purkinje neurons from wild-type mice in Fig 6A expected (IMPB1 staining in N2A cells in Fig 3B is mostly cytoplasmic)?

5. In Supplementary Fig 3, authors show significant increase in cell death in the cells from 30Q and 85Q groups but that have low GFP fluorescence i.e. are not transfected with any of these constructs? Can authors please explain this as there should be no 30Q or 85Q in these cells to cause cell death?

Reviewer #2 (Remarks to the Author):

I am sorry to say that some but not all comments have been dealt with by the authors during the revisions of the manuscript. In the opinion of this reviewer, major issues remain. The following comments have not been adequately addressed:

Response to R1 C/Q1 & MC/Q4:

It is nice that the authors included in vivo data to strengthen their claims. The new Figure 4 shows differences in importin beta1 localization upon expression of ataxin mutants. This does however not at all address the original comment that was rather about the fact that the authors have not convincingly shown re-localization of Nup98 because their construct does not appropriately localize in untreated tissue culture cells from the first place! This inconsistency is now even more obvious with the in vivo data show in Figure 9 were Ab staining was used (this is how Nup98 localization should look; compare localization observed Figures 7b, 8a-b to Figure 9)!

The images shown in Figure 9 do not show a convincing re-localization. The respective legend is quite nebulous about how exactly the image quantification has been done.

Response to R2 C/Q2:

I respectfully disagree that the in vivo data reinforce the in vitro data, because the localization of tested protein targets are not the same in cell culture and animals. The original comment that the different constructs do not show a consistent localization is again not at all addressed.

Response to R2 C/Q3:

The fact that the MS data are publicly available and will be subject of yet another manuscript is nice, but it does not address the concern that the author's analysis is qualitative and not quantitative, which is just not state of the art. Furthermore, the chosen controls are not very suitable, i.e. interaction studies of many nuclear proteins when used as bait might identify the nuclear transport machinery is abundant interactors and naturally biotinylated proteins will contaminate the BioID analysis.

Overall, although I am not convinced, I do not want to hold this manuscript back. So I would be fine with its publication if the other reviewers are.

Reviewer #3 (Remarks to the Author):

We are satisfied by the comments and manuscript changes offered.

Response to NCOMMS-17-19917A Reviewers' comments

Reviewer #1 (Remarks to the Author):

Authors have addressed several of the concerns improving the revised manuscript. Most importantly they included analysis of transgenic SCA1 model and human wt ATXN1[30Q] as a control, as well as more quantification of their results.

While these changes improved the manuscript, authors have also significantly decreased the importance/novelty of the manuscript by already publishing some of the results including the ataxin-1 interactome in their recent paper (Zhang et al, Sci Data 5, 180262, 2018).

RESPONSE 1. As noted in the Editor's email overview accompanying the publication decision, the importance/novelty has not been significantly decreased by the publication in the *Nature Communications* sister journal *Scientific Data* "we do not share Reviewer#1's concerns on importance/novelty." However, we also note that with the additional data included in this revised version (see 1.1), we have condensed the information on the interactome identification and analyses to present these as revised Figure 1, with pipelines shown in revised Figure S3). Our emphasis has also been revised in the Abstract ("Following identification of direct and proximal partners...."), and in the Introduction on page 2 to "analyzing" the interactome.

In addition there are several major concerns that need to be addressed.

1.1. Authors need to validate their interactome data, for at least some of their top five interactors (e.g. from Supplementary table 3). For example they could test ATXN1 binding of the identified ataxin-1 interacting partners through Co-IPs from the brain extracts.

RESPONSE 1.1.

We have addressed this issue specifically by with new analyses (revised Figure 2).

We have directed our efforts to confirmation of interactions in living cells, creating and optimizing a quantitative evaluation of interactions via the biomolecular fluorescence complementation (BiFC) approach. Importantly, this approach reports interactions in intact cells and avoids possible complications of extraction conditions and physical separation techniques that may less effectively capture proximal interactions.

Our results present robust positive and negative controls (Figure 2A), and robust CIC-ataxin-1 interaction, thus confirming the well-established interacting pair. Importantly, a reported mutant of the CIC interaction interface with ataxin-1 shows significantly reduced signal in our experiments (Figure 2B upper panels).

The validation of the ataxin-1 interaction with importin-alpha2 is not only presented in Figure 2B (lower panels) and Figure 2C (upper panels), but we have gone further to explore the nature of the interaction by our evaluation of the ataxin-1 NLS mutant, as well as 2 mutants of importin-alpha2 that modify interactions with NLS-cargo proteins (Figure 2C lower panels). Excitingly, we also show that the ataxin-1 mutant with abrogated interaction with CIC also is impaired in its interaction with importin-alpha2, thus providing the explanation to Reviewer 1's point 1.3 below.

These new results are described in the revised manuscript (pages 6-7), the analysis protocols are described (page 21), and we have included Mr A Lee as a co-author due his valuable contributions in creating the new constructs for these analyses (pages 15), performing the analyses and conducting 3 independent replicates of all presented data (see Author Contributions page 24).

1.2. Authors have found that overexpressing non-pathogenic WT ATXN1[30Q] in non-differentiated N2A cells causes similar disruption of nuclear transport as does overexpression of pathogenic expanded ATXN1[85Q]. Authors conclude that wild-type ATXN1[30Q] is toxic

and that “Ataxin-1 protein levels and cellular toxicity are correlated”, thus just increasing ATXN1 expression should cause these disruptions in nuclear transport. To support this they state that overexpression of wt ATXN1 can cause neurodegeneration (citing Gennarrino et al 2015 that showed that Pumilio haploinsufficiency upregulates ATXN1 expression (using Pumilio +/- mice), and causes cerebellar neurodegeneration). It is important for authors to note that in addition to increased ATXN1 expression, Pumilio +/- mice may have other, non-ATXN1 related features. Importantly, they should note that AO2 mice that overexpress ATXN1[30Q], are indistinguishable on rotarod from wild-type controls at 1 year of age (Clark et al. J Neuroscience 1997). Finally, to directly correlate nuclear transport disruptions to ATXN1 pathogenicity in vivo, authors should examine whether nuclear transport is disrupted in AO2 mice.

RESPONSE 1.2.

That higher levels of wild-type ataxin-1 can induce SCA1-like symptoms is well established in flies, cells and mouse models, and we have ensured that this revised manuscript cites additional supporting references – Ref 47. Fernandez-Funez et al., Nature, 2000 & Ref 48. Park et al, Nature 2013). We completely agree that there are likely to be many additional non-ataxin-1 related features in the Pumilio +/- mice, as many additional proteins/pathways will be impacted by the Pumilio haploinsufficiency, and now make this clearer on page 5 of the revised manuscript:

“In the scenario of Pumilio haploinsufficiency, even though many other targets are likely also affected, the upregulated wild-type ataxin-1 levels can lead to cerebellar neurodegeneration, and this could be largely corrected by reducing ataxin-1 levels ⁴⁵.”

However, the suggestion to include AO2 mice as studied in Clark et al 1997, does not address the complete phenotype described in that study. Whereas Reviewer 1 suggests these are a model showing no pathology, (i.e. “indistinguishable ... from wild-type controls at 1 year of age”), this ignores additional important phenotypic information provided in Clark et al:

- for the rotating rod test: “the rate of performance improvement seemed to be somewhat diminished in the A02/ transgenic mice. The absolute level of performance by A02/ animals on the rotating rod was less than that of wild-type mice on days 2–4”.
- for the bar cross test: “A02/1 transgenic mice performed similarly to B05/1 animals in all parameters scored except in Turns and Grooming Time, in which they had significantly lower levels of activity than wild-type animals (Fig. 3).”

We thus cite the work presented in Clark paper (now Reference #50) as part of the background to the study of the ataxin-1[30Q] interactome (on page 5) and again at the end of our Discussion regarding future advances (on page 13).

1.3. For their in vivo studies authors used transgenic mouse line ATXN1[82Q] that overexpresses mutant ATXN1 30-60 times. They also analyzed ATXN1[30Q]D776 mice that express phosphomimetic ATXN1[30Q] and demonstrate Purkinje neuron dysfunction, cerebellar pathology and ataxia, and ATXN1[82Q] V591A S602D mice expressing ATXN1[82Q] that cannot bind capicua and has significantly ameliorated phenotype. Intriguingly, authors claim that ATXN1[82Q] line demonstrates nuclear transport disruption (based on IMPB1 and NUP62), while ATXN1[30Q]D776 and ATXN1[82Q]V591A S602D do not. Since all three lines show accumulation of ATXN1 in the nucleus could this suggest that it is not overexpression of ATXN1 that causes these changes (as claimed above)? And if it is direct interaction of ATXN1 with nuclear transport proteins that causes disruption, authors should comment why is this abrogated when ATXN1 cannot bind Capicua?

RESPONSE 1.3.

The reviewer has raised an interesting issue, one that we have addressed via our interaction studies in the revised Figure 2 (as described in Response 1.1 above). Specifically, the ATXN1[82Q] V591A S602D mutant shows significantly less interaction also with importin-alpha2, thus further emphasizing the interesting relationships between the ataxin-1 protein

and the nuclear transport processes. We have highlighted this new relationship in our revised manuscript (pages 6 and 10).

1.4. Most of the results (e.g. Figs 3, 4, 5, 7, 8, SF2) are derived from two independent experiments. It is our opinion that to ensure reproducibility minimum of three independent experiments is required for any publication. Additionally, some of these figures are still lacking quantification (Fig 3, 7, 8, SF2).
RESPONSE 1.4.

All quantitative data has been obtained across multiple independent experiments, and now as described in the revised manuscript (see Methods: nuclear body size, Fn/c and Pearson's correlation analyses on pages 20-21; BiFC quantitation on page 21; Purkinje cell analyses on pages 21-22; analytical ultracentrifugation on pages 22- 23; cell death on page 19).

Thus, we have ensured that all microscopy data is now presented as typical images of the cells, as well as the quantitative analysis of images across the independent experiments. Thus, quantitative analysis has been added not only for all Fn/c measurements (Figs 4C, 5F&G, S6E, S10E), but as Pearson correlation coefficients (Figs 3E, 4B, 5E, S2B, S6D, S7E, S8B, S9C, S10D), for the BiFC interaction analyses (Fig 2A-C) and image analysis of changes in Purkinje cells in the transgenic animal models (Figs 6C, 7B & 7D). In addition the manuscript retains quantitative assessments of nuclear body size (Fig S1B), aggregate estimation by analytical ultracentrifugation (Fig S1D), cell death (Figs S4B & S5B).

1.5. Could authors please explain the use of the two different descriptive measures of nuclear transport disruptions in mice in Figs 6 (number or irregularity of the peaks) and 9 (shape of the nucleus)? It would be much more informative if authors would use the same quantifiable and objective measure for both figures. For example for both figs 6 and 9 authors could use % change in fluorescence from the rim to the inside of the nucleus?

RESPONSE 1.5.

We have clarified the different approaches on pages 21-22 of the revised manuscript, noting that whereas the nuclear rim fluorescence intensity changes for importin- β 1 in the SCA1 animals, it is the nuclear rim morphology (not fluorescence intensity) that alters for NUP staining in the SCA1 animals. We emphasise that all scoring was performed by an observer blinded to the animal genotype, and so is a robust analysis of observed changes.

6. With the exceptions of Figs 1 and 2, authors use untransfected cells as controls in most of their experiments. As transfection reagents, DNA, as well as overexpression of proteins, can all stress and alter cells, appropriate controls for those experiments with transfection and overexpression of MBI ATXN1 or GFP ATXN1 would be transfections with MBI or GFP vectors.

RESPONSE 1.6.

In all cell culture studies for analysis of the impact of ataxin-1, cells must be transfected to express the required ataxin-1 protein (Figures 3-5, S1-S2, S4-S10), either together with additional plasmids for the other protein of interest, or alone if an impact on the endogenous protein is analysed. Therefore, all cell studies are subjected to the interventions of transfection reagents, DNA and overexpression of proteins.

In Figure 1A, we included the negative control of GFP vector only. This shows the distribution of GFP throughout the nucleus and cytoplasm, and so recruitment into nuclear bodies requires analysis of GFP-ataxin-1 and quantitative analysis via Pearson's correlation coefficient calculations (Fig S8, Fig S9A/C).

In Figure 1C, we included the negative control of MBI vector only. This clearly shows the diffuse cytoplasmic distribution of the MBI-vector protein, without the formation of nuclear puncta (Fig 1C); the assessment of protein recruitment into nuclear bodies therefore uses MBI-ataxin-1.

Importantly, the MBI-ataxin-1[80Q] nuclear puncta do not recruit GFP only (Fig 3D and

quantitated in 3E), and this is the critical negative control for analyses in which we examine recruitment of GFP-tagged transporters/cargo/NUPs into MBI-ataxin-1[80Q] nuclear puncta (Fig 3A-C: importin-alpha2, importin-beta1, CRM1; Fig 4: NLS-beta-gal; Fig 5: importin-13, Hikeshi, NF-YB; Fig S6: importin-alpha2, importin-beta1, NLS-beta-gal; Fig S7: importin-alpha4, importin-7, importin-16, importin-5). Furthermore, for analyses of impact on cell death, and localization, GFP only is included as a control (Fig S4, and S5, respectively)

Minor issues:

1. The focus and depth of imaging can both influence nuclear rim staining. Could authors please describe imaging parameters they used to control for this?

RESPONSE 1.Minor 1.

For each assessment, we collected a 20 µm stack of images (1 µm/slice). We have included this detail in the Methods on page 22.

2. Can authors please comment on colocalization of GFP-NLS-B gal and MBI-ATXN1 in Fig 4A?

RESPONSE 1.Minor 2.

We have quantitatively analysed and confirmed co-localisation (Fig 4B and Fig S6D), noting this on page 8 and with the suggestion that this is “consistent with an importin-cargo interaction that is not dissociated within the ataxin-1 nuclear bodies.”

We also note that a similar phenomenon was observed for the cargoes NF-YB and HSP70, and so we have included the quantitative co-localisation analysis (Fig 5E) and noted this on page 9.

3. Could authors please describe how did they define cytoplasm for their Fn/c quantification?

RESPONSE 1.Minor 3.

We used CellProfiler cell image analysis software (version 2.1.1 for Mac) in which the nucleus is defined by DAPI staining, the total cell area is defined by the total fluorescence distribution in the cell, and the nucleus is defined by subtracting the nuclear area from the total. We have included this detail in the Methods on page 20/21.

In addition, we have undertaken additional quantitative analyses, and added the following explanations

- all imaging studies were performed on ≥ 3 independent occasions including preliminary optimization steps, then the acquired images were quantitatively analyzed using Fiji software (see Methods on page 20)
- defining our calculations of protein colocalisation/mis-localisation, according to the Pearson's correlation coefficients (see Methods on pages 20/ 21)
- reporting our quantitative analyses of the BiFC results (see Methods on page 21)

4. Is lack of cytoplasmic IMPB1 staining in Purkinje neurons from wild-type mice in Fig 6A expected (IMPB1 staining in N2A cells in Fig 3B is mostly cytoplasmic) ?

RESPONSE 1.Minor 4.

We tested a range of anti-importin or anti-NUP antibodies for their suitability for use in the tissue staining protocol, and as described on page 21, those chosen showed appropriate specificity and localisation patterns in control brain sections.

In both cell and animal systems, prominent nuclear rim localisation for IMPβ1 (GFP-IMPβ1 in the cells, anti-IMPβ1 staining in the mice) was noted and we have reworded those sections to ensure that the common features of localization are more clearly stated (see Results on pages 7-8 and page 10-11). That there is additional cytoplasmic localization for GFP-IMPβ1 has been acknowledged and is most likely attributed to the exogenous expression of GFP-IMPβ1; thus as stated on page 11 this may result from higher levels of expressed importin-β1.

5. In Supplementary Fig 3, authors show significant increase in cell death in the cells from 30Q and 85Q groups but that have low GFP fluorescence i.e. are not transfected with any of these constructs? Can authors please explain this as there should be no 30Q or 85Q in these cells to cause cell death?

RESPONSE 1.Minor 5.

This data is now presented in Figure S4 in the revised manuscript. We have corrected our gating to exclude cells recorded with no expressed GFP-tagged proteins, i.e. data is presented with gating from 0.3-1K. Thus, we maintain the conclusions on the impact of ataxin-1[30Q] and ataxin-1[85Q] as described on page 5.

Reviewer #2 (Remarks to the Author):

I am sorry to say that some but not all comments have been dealt with by the authors during the revisions of the manuscript.

RESPONSE2.i

We respond to the specific issues raised by this reviewer in the text below, but first we have returned to the first round of reviews to re-confirm what other comments might have been considered as overlooked in the previous revision. The following Reviewer comments are directly followed by our responses.

- *The authors base their analysis on two different construct: GFP-ataxin-1-[85Q] and MBI(myc- BirA)-ataxin-1-[85Q]. In Figure 1 A and C the pattern of these two constructs are shown but the staining pattern is clearly different. Detection of GFP shows distinct spots but anti-Myc has a diffuse localization all over the nucleus. The authors do not comment on these different patterns even though all further experiments are based on them. It should have been investigated how these differences can be explained. Does the fusion of GFP or Myc-BirA* influence the localization?*

We have now reworded our description of the comparison of GFP-ataxin-1 and MBI-ataxin-1 on pages 3-4 of the revised manuscript, highlighting the differences noted but also emphasizing that these two constructs are used in combination in our studies:

“In contrast to MBI alone that was primarily cytoplasmic, MBI-ataxin-1[85Q] was largely restricted to the nucleus (Figure 1C). This localization was consistent with the nuclear localisation of GFP-ataxin-1[85Q] and the previous observation that nuclear localisation of ataxin-1 is important for its toxicity³⁷, albeit that MBI-ataxin-1[85Q] formed less distinctive nuclear bodies than that observed for GFP-ataxin-1[85Q] (Figure 1C (anti-myc staining) vs Figure 1A (GFP signal)). Thus, to assess the suitability of MBI-ataxin-1[85Q] further, we used streptavidin-based detection to detect biotinylation driven by the MBI-fusion constructs. We confirmed a widespread distribution of biotinylation throughout the cell nucleus and cytoplasm by MBI alone, whereas we observed that the MBI-ataxin-1[85Q]-driven biotinylation was largely localised to prominent nuclear bodies (Figure 1C). Furthermore, we observed a more restricted biotinylation profile for MBI-ataxin-1[85Q] as defined by streptavidin-detection of biotinylated proteins separated by SDS-PAGE (Figure 1D), consistent with interactions of the MBI-ataxin-1[85Q] dominated by those in a nuclear body context. Thus, GFP-ataxin-1[85Q] and MBI-ataxin-1[85Q] have the potential to provide independent information on the nuclear-interacting proteins of ataxin-1[85Q].”

We further highlight on page 10 of the revised manuscript, the importance of studies in the SCA1 mouse models, as these results are independent of tagged ataxin-1:

“Importantly, our in vivo studies could be performed by analysing the localisation of endogenously expressed proteins, and so can provide an assessment independent of epitope-tag position or identity.”

- *Is there a difference when the fusion is done on the C-Terminus of ataxin-1[85Q]?*

For the GFP-ataxin-1 and MBI-ataxin-1 constructs used in our study, the fusion was achieved to the N-terminus of the ataxin-1 protein; notably most published studies have taken this approach to avoid possible interference by fusions to the C-terminus that may be in closer proximity to important signal and/or structure regions such as the NLS or AXH domain as reported previously (see review, Zoghbi HY, Orr HT. Pathogenic mechanisms of a polyglutamine-mediated neurodegenerative disease, spinocerebellar ataxia type 1. *J. Biol. Chem* **284**, 7425-7429 (2009)). However, even the fusion of a fluorescence tag such as DsRed2 to the C terminal of ataxin-1 has already been previously reported to not influence the formation of ataxin-1 NBs and its nuclear localization (Fujita et al., A functional deficiency of TERA/VCP/p97 contributes to impaired DNA repair in multiple polyglutamine diseases. *Nat Commun* **4**, 1816 (2013)). Furthermore, in our BIFC studies, the ataxin-1 is a fusion at its C-terminus to the Venus fragment, and as shown in Figure 2, this retains its punctate nuclear localization.

- *The immunofluorescence staining done in all further experiments are using either one of the construct. How were those choices made? Since already the initial experiment shows differences, all staining should be performed using both constructs.*

We consistently evaluated localisation of GFP-tagged transporters, as highlighted on page 15 of the revised manuscript:

“to assess the influence of ataxin-1 on nuclear transport in Neuro-2a cells, we assessed a range of nuclear proteins, choosing GFP-tagged proteins to ensure consistent comparisons when co-transfected with MBI-ataxin-1”

It is not possible to study GFP-ataxin-1 in the presence of GFP-transporters. Thus, rather than recloning all transporters into alternatively-tagged vectors, we directed our attention to extending our analyses to the accepted mouse model of SCA1 pathology. Importantly, and as noted above, the results obtained in the SCA1 animal models are independent of the tag identity.

In the opinion of this reviewer, major issues remain.

The following comments have not been adequately addressed:

2.1 Response to R1 C/Q1 & MC/Q4: It is nice that the authors included in vivo data to strengthen their claims. The new Figure 4 shows differences in importin beta1 localization upon expression of ataxin mutants. This does however not at all address the original comment that was rather about the fact that the authors have not convincingly shown re-localization of Nup98 because their construct does not appropriately localize in untreated tissue culture cells from the first place! This inconsistency is now even more obvious with the in vivo data show in Figure 9 where Ab staining was used (this is how Nup98 localization should look; compare localization observed Figures 7b, 8a-b to Figure 9)!

The images shown in Figure 9 do not show a convincing re-localization. The respective legend is quite nebulous about how exactly the image quantification has been done.

Response to R2 C/Q2: I respectfully disagree that the in vivo data reinforce the in vitro data, because the localization or tested protein targets are not the same in cell culture and animals. The original comment that the different constructs do not show a consistent localization is again not at all addressed.

RESPONSE 2.1.

Although the Reviewer has re-voiced concerns regarding NUP localization patterns, we stress that our results were consistent with published findings (e.g. for Nup98: Griffis et al., 2002 *Mol Biol Cell* **13**, 1282-1297 & *Mol Biol Cell* **14**, 600-610 (2003); for Nup62: Kinoshita et al., *PLoS one* **7**, e36137 (2012)). However, in revising the manuscript for resubmission, we have now removed the cell-based Nup staining or overexpression data (previous figures 7, 8 & supp figures S11, and also the further cell studies following up those observations in

previous supp figures S10, S12, S13), thus stream-lining the transition from the Imp-staining in the brains of the SCA1 mice to the Nup-staining also in those animals. These changes do not change the major conclusions of our study, and indeed the simplification should facilitate a clearer understanding of the important in vivo results.

For the in vivo staining results, we have clarified the legend to the Figure (revised Figure 7, page 34). We also have revised the description of the results (pages 11-12) to clarify how these images were scored by an observer blinded to the animal phenotype:

“We noted disruption to the nuclear rim morphologies as detected by staining for either NUP98 or NUP62, but without noticeable relocalisation to the ataxin-1 nuclear bodies (Figure 7A & C). Further quantitative analysis (50-100 cells/animal, 200-350 cells total) by an observer blinded to the animal genotype, and defining categories of either normal-mild (nuclear rim staining round to mostly round with ≤ 2 small to medium sized wrinkles) or moderate-severe (nuclear rim staining observed to be more elongated with large invaginations and distorted morphology) confirmed disruption in the *ATXN1*[82Q] but not *ATXN1*[30Q]S776D and *ATXN1*[82Q]CIC(mVS) animals (Figure 7B & D).”

2.2 Response to R2 C/Q3: The fact that the MS data are publicly available and will be subject of yet another manuscript is nice, but it does not address the concern that the author's analysis is qualitative and not quantitative, which is just not state of the art. Furthermore, the chosen controls are not very suitable, i.e. interaction studies of many nuclear proteins when used as bait might identify the nuclear transport machinery is abundant interactors and naturally biotinylated proteins will contaminate the BioID analysis. Overall, although I am not convinced, I do not want to hold this manuscript back. So I would be fine with its publication if the other reviewers are.

RESPONSE 2.2.

We acknowledge the value of quantitative analyses, and have thus modified the manuscript regarding the limitations of our current analyses as well as the importance of future quantitative analysis as follows.

- On page 4, we have ensured that it is clear that our analyses are qualitative, and that we have chosen to integrate two qualitative approaches.
- On page 4, we reiterated our approach to removal of background proteins.
- On page 4, the advantage of quantitative proteomics including labeling and multiplexed analysis is acknowledged.
- On pages 5, the use of Ingenuity Pathway Analysis is described to statistically test for pathways showing over-representation relative to that expected by chance.
- On page 13, we conclude with highlighting the importance of quantitative proteomic analyses

Reviewer #3 (Remarks to the Author):

We are satisfied by the comments and manuscript changes offered.

Reviewers' comments:

Reviewer #1 (Remarks to the Author):

Authors have addressed most of the concerns, including the validation for two of the partners and clarifications of the analysis. While most of these changes improved the manuscript, some raised additional questions, and some remain unresolved.

1. Response to 1.1. Regarding confirmation of the interaction between capicua and importin α 2 in Fig 2:

1. As expression level of each interactor can lead to the change in the biomolecular fluorescence complementation (BiFC), authors need to show that 85Q, 85Q (mV5), 85Q K772T and interactors CIC and importin have similar expression in all different conditions.

2. Fig 2 legend states "Results represent the mean \pm SEM for a single typical experiment". It is important that the quantified data presented in histograms is a mean of all the performed independent experiments.

In addition authors should state how many cells were quantified, on how many coverslips, and include exact number of independent experiments that were used to derive data.

2. Response to 1.3. Authors did not clarify/ discuss why ATXN1[30Q]D776 mice do not show changes in nuclear transport that ATXN1[82Q] mice do. This is confusing since authors show that overexpressing ATXN1[30Q] and ATXN1[82Q] similarly affect nuclear transport in N2A cells (Figs 3 and 4 and Supplementary figure S6).

3. Response 1.4. Please see 1.2 above. In addition in method section authors state "All imaging studies were performed on ≥ 3 independent occasions including preliminary optimization steps". Unless optimization steps are done exactly the same way as final experiments, I respectfully disagree that the "preliminary optimization steps" experiments can be used as independent replicate of final experiments. The exact number of times that experiments were independently performed (different dates) should be included with each experiment. In addition, for the experiments using mice authors should clearly state what is presented-e.g. for Figs 6 and 7 is it mean % of cells from n=3 mice, of mean % of cells from N= ? sections, and how many sections did they analyzed. It would be helpful to include what kind of statistical test was used to derive p values in each figure legend.

4. Minor 5 Response. It is still a mystery why is there significant increase in cell death in untransfected cells (low GFP intensity) derived from ATXN1[30Q], or ATXN1[82Q] transfection wells vs GFP wells. None express any protein.

Additional issues that arose:

1. Authors state that increased size of nuclear bodies with arsenite in GFP-ATXN1[85Q] expressing cells, that was not observed in GFP-ATXN1[85Q]S776A cells, paralleled increases in Atxn1 aggregate formation as assessed by analytical ultracentrifugation (Supplementary Fig 1D). However, Figure S1D seems to indicate that arsenite induces aggregation in S776A cells (from 0 to 31.7). Could authors please clarify this contradiction?

2. Nucleolin was identified as MBI-ataxin-1[85Q] and [30Q] interactor (Supplementary table 3). So it is surprising that there is no co-localization between nucleolin and MBI-Ataxin1 [85Q] in Suppl Fig 2. Could authors clarify/discuss this?

3. Could authors clarify difference in ATXN1 induced death in different figures: in S4 ATXN1[85Q] causes ~ 30% cell death, while in S5 it is ~ 8%?

4. Figs S5B and C are missing, and S5B is actually S5D based on the figure legend.

5. Could authors clarify/discuss opposing effects on ATXN1[85Q] nuclear transport alterations on different cargo proteins—for example GFP-NLS-Bgal becomes more cytoplasmic while NF-YB and HSP7 become more nuclear (Figs 4 and 5)?

Responses to Reviewer 1

We thank the reviewer for his/her time in making detailed comments on our revised manuscript; we address each of the suggestions below.

1. Response to 1.1. Regarding confirmation of the interaction between capicua and importin α 2 in Fig 2:

1. As expression level of each interactor can lead to the change in the biomolecular fluorescence complementation (BiFC), authors need to show that 85Q, 85Q (mV5), 85Q K772T and interactors CIC and importin have similar expression in all different conditions.

RESPONSE:

Reviewer #1 appears to be requesting controls for these BiFC experiments. We reiterate that multiple previously established controls are included in our rigorous analyses. Moreover, these BiFC results are not an end-point of our analyses, but rather provide one step in our scientific argument, not only supporting our proteomics analyses, but also, in turn, confirmed by our extensive subsequent *in vitro* and *in vivo* analyses reinforcing the importance of altered nuclear transport.

Specifically, we validated our BiFC approach by our inclusion of multiple, established negative and positive controls, i.e. we specifically set out experimental results in Figure 2 with panels A and B confirming the expected interactions of c-Fos/c-Jun and ataxin-1[85Q]/CIC dependent on specific domains as previously reported. From our understanding of previously published studies using BiFC analyses, these are the gold standard controls; the Reviewer does not provide any referenced arguments to the contrary.

We respectfully disagree that assessment of protein levels could provide additional robust information to enhance the manuscript. Specifically, protein-protein interactions are governed by the affinities of the interaction; these are reported as dissociation constants (K_D). Importantly, according to the biophysical principles dictating protein-protein interactions, when concentrations exceed the K_D value, interactions are saturated and independent of concentrations of the interacting partner.

Despite this, and to satisfy the Reviewer, we now further clarify the importance of the controls that we included in our analyses, adding the following to page 22 of the Methods:

“For the assessment of ataxin-1[85Q] association with selected binding partners in cells, we exploited the biomolecular fluorescence complementation (BiFC) system that is based on protein interactions bringing together ectopically expressed Venus N-terminal (VN-) and Venus C-terminal (VC-) fragments to reconstitute fluorescence of the Venus protein⁵⁴. The BiFC system enables detection of interactions with minimal perturbation to the cells, requires no assumptions about the accessibility of the complex to extrinsic fluorophores, and is sufficiently sensitive to enable analysis of interactions between proteins expressed at levels comparable to endogenous proteins⁵⁵. Importantly, BiFC analysis allows the detection of interactions that may involve only a subpopulation of a particular protein, which can specifically interact with many cellular proteins⁵⁷.”

Thus, we cultured HeLa cells on coverslips, transfected to express the indicated BiFC constructs (VN- and VC-), and stained with 0.8 μ g/ml Hoechst 33342 (Sigma Aldrich) prior to imaging by confocal laser microscopy (FluoView™ FV1000 confocal microscope with 60 \times water immersion objective). To ensure our system robustly reported intracellular protein-protein interactions for high levels of ectopically expressed proteins, we included the controls of the interaction of c-Fos

and c-Jun and its disruption for the mutant c-Fos(⊗zip)⁸⁹, and the interaction of CIC and ataxin-1[85Q] and its disruption for the mutant ataxin-1[85Q] mVS⁵⁸, as well testing the interactions between wild-type, mVS mutant, or the nuclear localisation defective K772T mutant ataxin-1[85Q] with Impα2 expressed as either wild-type, an NLS-binding deficient mutant⁵⁹ (importin-α2 D192K/E396R, abbreviated herein as importin-α2 mDE) or a mutant lacking the importin-β-binding domain within its N-terminus⁶⁰ (importin-α2 with residues 70-529 deleted, abbreviated herein as importin-α2 (⊗IBB)). Quantitative analysis of the Venus fluorescence intensity reconstituted by molecular complementation was performed (ImageJ 1.52n public domain software) each in 3 independent experiments. In these analyses, nuclei were defined by Hoechst channel using the threshold command. Nuclear fluorescence, corrected for background in the absence of BiFC expression, was then determined for 80-200 cells/sample.”

Furthermore, we provide additional clarification in the text where the BiFC assay is introduced (page 6 of the manuscript), so that the system and its controls are more clearly described:

“We established the robustness of these identifications of partners for the ataxin-1[85Q] protein by employing a bimolecular fluorescence complementation (BiFC) assay that relies on protein interactions bringing together ectopically expressed Venus N-terminal (VN-) and Venus C-terminal (VC-) fragments to reconstitute fluorescence of the Venus protein, thus allowing direct visualization of protein interactions in their normal cellular environment⁵⁴. Importantly, BiFC analysis also allows detection of weak and transient interactions^{55,56}. We first confirmed the robustness of this system by demonstrating that the known interacting pair of transcription factors, c-Fos and c-Jun, resulted in strong and homogeneous nuclear fluorescence consistent with previous studies⁵⁷, thus providing a robust positive control for this assay system; notably, in parallel testing of the c-Fos mutant, c-Fos⊗zip, paired with c-Jun, quantitative analysis confirmed nuclear fluorescence was significantly decreased consistent with the previous studies⁵⁷ and thus providing a robust negative control for this assay system (Figure 2A).

Next, we used the interaction of ataxin-1[85Q] and the previously reported partner, CIC, as a further positive control for the BiFC system, as well as also testing the binding between ataxin-1[85Q] and our newly identified partner from the nuclear transport pathway, importin-α2. “

2. Fig 2 legend states “ Results represent the mean ± SEM for a single typical experiment”. It is important that the quantified data presented in histograms is a mean of all the performed independent experiments. In addition authors should state how many cells were quantified, on how many coverslips, and include exact number of independent experiments that were used to derive data.

RESPONSE:

We have followed the Reviewer’s suggestion, and modified our Figure 2 and its Figure legend accordingly. Furthermore, all data graphs are now presented with individual data points (rather than bar graphs) as requested by the Editor. A spreadsheet with all source data is now also provided, as requested by the Editor.

2. Response to 1.3. Authors did not clarify/ discuss why ATXN1[30Q]D776 mice do not show changes in nuclear transport that ATXN1[82Q] mice do. This is confusing since authors show that overexpressing ATXN1[30Q] and ATXN1[82Q] similarly affect nuclear transport in N2A cells (Figs 3 and 4 and Supplementary figure S6).

RESPONSE:

We hypothesize that nuclear transport/pore changes are correlated with the progressive disease seen in ATXN1[82Q] as opposed to early dysfunction seen in ATXN1[30Q]D776. We do not see Purkinje cell loss in ATXN1[30Q]D776 compared to ATXN1[82Q] mice.

To satisfy the Reviewer, we now extend our discussion in the manuscript as follows.

Page 12:

“Interestingly, these observations reinforce the links between end-stage SCA1 pathology and altered localization of nuclear transport proteins as exemplified by importin- β 1, i.e. the disruption of nuclear transport proteins in the in vivo setting appears to be more closely aligned with the progressive disease to late stage dysfunction observed in *ATXN1*[82Q] animals rather than the features of early dysfunction that do not proceed to Purkinje cell loss in the *ATXN1*[30Q]S776D animals.”

Page 13:

“Thus, the disruption of NUP98 and NUP62 staining patterns only in the *ATXN1*[82Q] transgenic mice leads to the suggestion that nuclear pore changes are of highest relevance in progressive SCA1 pathology but likely to be less critical in the early dysfunction seen in the *ATXN1*[30Q]S776D model in vivo despite our observations that acute overexpression of ataxin- 1[30Q] or ataxin-1[85Q] in vitro can initiate comparable toxicity and changes in nuclear transport. Thus, the translation of findings from the scenario of in vitro cell biology to in vivo testing in animal models is critically important in understanding the mechanisms contributing to long term pathology.”

3. Response 1.4. Please see 1.2 above. In addition in method section authors state “ All imaging studies were performed on ≥ 3 independent occasions including preliminary optimization steps”. Unless optimization steps are done exactly the same way as final experiments, I respectfully disagree that the “preliminary optimization steps” experiments can be used as independent replicate of final experiments. The exact number of times that experiments were independently performed (different dates) should be included with each experiment. In addition, for the experiments using mice authors should clearly state what is presented-e.g. for Figs 6 and 7 is it mean % of cells from $n=3$ mice, of mean % of cells from $N=?$ sections, and how many sections did they analyzed. It would be helpful to include what kind of statistical test was used to derive p values in each figure legend.

RESPONSE:

We now include the details as required – the number of independent replicate experiments, or animals, in all Figure legends as appropriate. In all figure legends we additionally include the precise number of cells analysed and the statistical test used for further analysis. All source data has also now been provided in the spreadsheet as requested by the Editor.

4. Minor 5 Response. It is still a mystery why is there significant increase in cell death in untransfected cells (low GFP intensity) derived from ATXN1[30Q], or ATXN1[82Q]transfection wells vs GFP wells. None express any protein.

RESPONSE:

According to our gating protocol, cells not expressing proteins are gated as 0-0.3K. Thus, the cells in the 0.3-1K range express protein, and show impact of that expressed protein.

We have clarified this as follows.

In the methods (page 20):

“Fluorescence detection and recording was assessed by flow cytometry (BD LSRFortessa (BD Bioscience)) with the background fluorescence control (i.e. fluorescence counts from the green channel = 0 - 0.3K) estimated from using untransfected cells and negative controls determined by the GFP-only cells (GFP, Green channel) and SYTOX-only cells (SYTOX, Red channel).”

In the legend to Figure S4 (page 40/41):

“Non-expression of GFP proteins was gated as 0-0.3K and three ranges of GFP fluorescence counts (x-axis: FITC-A) were gated and analysed: 0.3-1K, 1-10K, 10-100K. In each GFP fluorescence range, SYTOX Red was gated (y-axis: APC-A) for live cells or dead cells as indicated.

(B) For each range of GFP fluorescent counts, the % cells within the population were estimated.

(C) Cell death (%) was calculated for cells with the different levels of GFP, GFP-ataxin-1[30Q], or GFP-ataxin-1[85Q] detection to allow estimates of toxicity at low (0.3-1K), moderate (1-10K) and high (10-100K) levels of GFP-tagged protein expression. Results are calculated from 3 independent experiments.

Additional issues that arose:

1. Authors state that increased size of nuclear bodies with arsenite in GFP-ATXN1[85Q] expressing cells, that was not observed in GFP-ATXN1[85Q]S776A cells, paralleled increases in Atxn1 aggregate formation as assessed by analytical ultracentrifugation (Supplementary Fig 1D). However, Figure S1D seems to indicate that arsenite induces aggregation in S776A cells (from 0 to 31.7). Could authors please clarify this contradiction?

RESPONSE:

We have expanded the first paragraph (page 3) describing our initial observations of the ataxin-1 nuclear bodies to clarify as follows:

“Notably, our assessment of ataxin-1 aggregate formation by analytical ultracentrifugation analyses of lysates prepared from these cells, showed increased aggregates for both GFP-ataxin-1[85Q] and the non-pathogenic GFP-ataxin-1[85Q] S766A (Figure S1D), emphasizing that aggregation state and visible body size do not always change in parallel as has been reported for the huntingtin protein with an expanded polyQ tract^{36,37} and as we have recently observed for polyQ-ataxin-1³⁸.”

2. Nucleolin was identified as MBI-ataxin-1[85Q] and [30Q] interactor (Supplementary table 3). So it is surprising that there is no co-localization between nucleolin and MBI-Ataxin1 [85Q] in Suppl Fig 2. Could authors clarify/discuss this?

RESPONSE:

Nucleolin was only identified in the BioID protocol, and thus is expected as only a proximal partner which does not guarantee a sustained interaction and co-localisation. This underlies our choice to include proteins only identified in 3 of the 4 conditions, and we have included this further in our manuscript on page 5 as follows:

“In analysing the ataxin-1[85Q] interactome, we further refined the list of 675 proteins. Noting that the nucleolin protein was identified only by the BioID approach and thus may not show sustained interaction or co-localisation with ataxin-1[85Q], and that our initial characterisation clearly distinguished nucleoli from ataxin-1 nuclear bodies (Figure S2), we included only the 91

proteins observed in at least 3 out of 4 of the conditions analysed (Figure 1E, indicated by *), thus ensuring their identification by both BioID and pull-down approaches.”

3. Could authors clarify difference in ATXN1 induced death in different figures: in S4 ATXN1[85Q] causes ~ 30% cell death, while in S5 it is ~ 8%?

RESPONSE:

Here, the Reviewer highlights the cell death only in the 10-100K population in Figure S4. Our results for cell death of the 0.3-1K and 1-10K populations, that are the majority of the transfected cells, are actually in good agreement with the results in Figure S5.

To avoid this potential confusion for the reader, we now include population analyses for the different expression levels as Figure S4B. The calculated death % in these subsets is now presented in Figure S4C. We further clarify these data addressing Figure S4 on page 5 of the revised manuscript as follows:

“We initially confirmed that comparable levels of ataxin-1[30Q] and ataxin-1[85Q] could cause comparable increases in cell death (Figure S4), stratifying our population analysis for levels of protein expression. Specifically we noted that, in the majority of the cell population expressing either low (0.3-1K GFP fluorescence count) or moderate (1-10K GFP fluorescence count) ataxin-1 proteins, cell death was significantly increased to ~8%, whereas higher levels (10-100K GFP fluorescence count) in a smaller subset of the cell population (~7%) could further cause significant increases in death to ~30% of that population subset.”

This is further explained in the methods (page 20):

“For analysis to compare toxicity of GFP-ataxin-1[30Q] with GFP-ataxin-1[85Q] expressed at different levels within the cell population, further gating was applied to the transfected cells based on different fluorescence counts from the green channel: 0.3-1K, 1-10K, 10-100K. Cell death percentage (%) in each fluorescence range was determined using the following formula: (number of dead cells/number of total cells) x 100.”

Furthermore, the legend to Figure S4 (page 40/41) directly addresses this issue of levels:

“Cell death (%) was calculated for cells with the different levels of GFP, GFP-ataxin-1[30Q], or GFP-ataxin-1[85Q] detection to allow estimates of toxicity at low (0.3-1K), moderate (1-10K) and high (10-100K) levels of GFP-tagged protein expression. Results are calculated from 3 independent experiments.”

For Figure S5, we emphasize on page 7 that our estimation of cell death was taken across the population (not stratified for expression level as per Figure S4):

“Notably, toxicity as assessed across the total population of GFP-protein expressing cells was significantly lower for GFP-ataxin-1[85Q]K772T than for wild-type GFP-ataxin-1[85Q] (Figure S5B&C).”

Thus, there is good agreement between the transfected population analysis (8% cell death, Figure S5) and the majority of the population (8% cell death in the 0.3-1 and 10K populations, Figure S4).

4. Figs S5B and C are missing, and S5B is actually S5D based on the figure legend.

RESPONSE:

We thank the reviewer for alerting us to this error in the Figure legend. The legend has now been

corrected.

5. Could authors clarify/discuss opposing effects on ATXN1[85Q] nuclear transport alterations on different cargo proteins –for example GFP-NLS-Bgal becomes more cytoplasmic while NF-YB and HSP7 become more nuclear (Figs 4 and 5)?

RESPONSE:

We have now more specifically noted the changes in importins and the exportin CRM1 on page 9 as follows:

“Mis-localisation of nuclear transporters could be expected to result in altered nucleocytoplasmic distribution of their cargo proteins, albeit that our observations that multiple importins as well as the exportin CRM1 are impacted in the presence of ataxin-1[85Q] (Figures 3 and S6) would complicate predictions of the magnitude or direction of the change to favour/disfavour nuclear import or export and the resulting change in nuclear to cytoplasmic ratio (Fn/c).”

We also reiterate this issue on page 10 as follows:

“Thus, in the presence of ataxin-1[85Q], the changes in importins as well as CRM1 have the capacity to reduce the nuclear accumulation of some proteins (as exemplified by GFP-NLSb-gal, Figure 4) whilst increasing nuclear accumulation of other proteins (as exemplified by NF-YB and HSP70, Figure 5).”